# Synthesis and macrocyclization-induced emission enhancement of benzothiadiazole-based macrocycle

Shuo Li[1], Kun Liu[2], Xue-Chen Feng[2], Zhao-Xian Li[2], Zhi-Yuan Zhang[2], Bin Wang [2], Minjie Li [1], Yue-Ling Bai[1 ✉], Lei Cui[1 ✉] & Chunju Li [1,2 ✉]

We presented an effective and universal strategy for the improvement of luminophore's solid-state emission, i.e., macrocyclization-induced emission enhancement (MIEE), by linking luminophores through $C(sp^3)$ bridges to give a macrocycle. Benzothiadiazole-based macrocycle (BT-LC) has been synthesized by a one-step condensation of the monomer 4,7-bis(2,4-dimethoxyphenyl)−2,1,3-benzothiadiazole (BT-M) with paraformaldehyde, catalyzed by Lewis acid. In comparison with the monomer, macrocycle BT-LC produces much more intense fluorescence in the solid state ($\Phi_{PL} = 99\%$) and exhibits better device performance in the application of OLEDs. Single-crystal analysis and theoretical simulations reveal that the monomer can return to the ground state through a minimum energy crossing point ($MECP_{S1/S0}$), resulting in the decrease of fluorescence efficiency. For the macrocycle, its inherent structural rigidity prohibits this non-radiative relaxation process and promotes the radiative relaxation, therefore emitting intense fluorescence. More significantly, MIEE strategy has good universality that several macrocycles with different luminophores also display emission improvement.

[1] College of Sciences, Center for Supramolecular Chemistry and Catalysis, Shanghai University, Shanghai 200444, People's Republic of China. [2] Tianjin Key Laboratory of Structure and Performance for Functional Molecules, College of Chemistry, Tianjin Normal University, Tianjin 300387, People's Republic of China. ✉email: yuelingbai@shu.edu.cn; cuilei@shu.edu.cn; cjli@shu.edu.cn

Organic luminescent materials with high quantum efficiencies have attracted intensive attention due to their extensive applications in sensors[1–3], bioimaging[4–7], laser displays[8–10], light-emitting diodes[11–14], and anti-counterfeiting[15,16]. However, most of the organic luminogens suffer from a severe quenching effect in the aggregate state due to the formation of such detrimental aggregates as excimers and exciplexes[17–19], which greatly limits their applications in organic luminescent materials. To address aggregation-causing quenching (ACQ) issue, some effective methods have been built to improve emission efficiency, for example, aggregation induced emission (AIE)[20–26], crystallization induced emission (CIE)[27–29], and supramolecular assembly induced emission enhancement[30,31]. In these cases, it is highly dependent on the restriction of intramolecular motions and control of the twisted conformation of organic luminophores in the solid state[32,33]. It is still urgent to develop a new strategy for emission enhancement, which would be not only helpful to the construction of fantastic fluorophores and materials, but also significant to understand the relationship between luminescent mechanism and molecular structures.

In the past ten years, our group has focused on the synthesis and applications of new macrocycles, and has developed a versatile methodology for functional biphen[n]arenes[34]. We predicted that the macrocyclization of organic luminophore through $sp^3$ methylene, i.e., the construction of luminophore-based macrocyclic arene, would efficiently enhance the emission. Such emission enhancement is theoretically feasible considering the following two features: on one hand, it spatially separates chromophores in a single macrocycle to eliminate the concentration quenching to a certain degree[35–38], on the other hand, it restricts intramolecular motion by locking its chromophores into the skeleton of macrocycle to suppress non-radiation relaxation[39,40]. Herein, we report the synthesis of a benzothiadiazole-based macrocycle (BT-LC) with three methylene bridges, which exhibits high fluorescence quantum yield in the solid state, up to 99%, much higher than that of BT-M. Since the emission enhancement is due to the cyclization of a few of luminogens by methylenes, it is termed as macrocyclization-induced emission enhancement (MIEE) (Supplementary Scheme 1). Experiments and theoretical calculations demonstrated that the enhanced emission can be ascribed to the efficient suppression of non-radiative relaxation process. For the application of organic light-emitting diodes (OLEDs), the device containing the macrocycle exhibits higher maximum brightness ($B_{max}$) and external quantum efficiency ($EQE_{max}$) than that for the monomer. MIEE reported here would be a general strategy for improving emission efficiency, and has the potential to be practically utilized in organic luminescent materials.

## Results

**Synthesis of BT-LC.** The synthetic route of BT-LC is illustrated in Fig. 1. The precursor 4,7-bis(2,4-dimethoxyphenyl)−2, 1,3-benzothiadiazole (BT-M) was synthesized through Suzuki–Miyaura coupling reaction of 4,7-dibromo-2,1,3-benzothiadiazole and 2,4-dimethoxybenzeneboronic acid. 2,1,3-benzothiadiazole is a widely used building block in luminescent materials[41,42]. BT-M is an ideal highly-emissive molecule, due to its D-A architecture with dimethoxyphenyl donor and 2,1,3-benzothiadiazole (BT) acceptor, where the push-pull system can improve the luminescence performance through the enhancement of intramolecular charge transfer (ICT)[43–45]. Subsequently, BT-LC was synthesized by a one-step condensation of BT-M with paraformaldehyde, catalyzed by Lewis acid of $BF_3 \cdot Et_2O$ (52% yield). No other cyclic oligomers such as tetramer and pentamer were observed. All chemical structures were confirmed by $^1H$ and $^{13}C$ NMR spectroscopy, high resolution mass spectra (HRMS) as well as single crystal X-ray diffraction (Supplementary Figs. 2–7 and 45–50).

**Photophysical properties of BT-LC.** As depicted in Fig. 2, BT-LC exhibited a red-shifted emission ($\lambda_{em} = 562$ nm) compared to BT-M ($\lambda_{em} = 491$ nm). Also, enhanced fluorescence was observed in the photoluminescence (PL) spectra. The quantum yield ($\Phi_{PL}$) for BT-LC (99%) is much higher than that for the monomer (65%). The time-resolved emission decay properties of BT-LC and BT-M in the solid state were also studied. BT-LC gave relatively long fluorescence lifetime (11.25 ns), in comparison with BT-M (8.45 ns) (Supplementary Figs. 36, 37). BT-LC's absorptions show negligible differences with $\lambda_{abs}$ values of 401–410 nm, and its emissions show remarkable bathochromic shifts as the solvent polarity increases (Supplementary Fig 32). This is due to ICT effect resulted by their distinguished D–A architectures[46–48]. Besides in the solid state, BT-LC also exhibits high $\Phi_{PL}$ values of 83–89% in solution (Supplementary Table 1). Moreover, BT-LC shows dual-state emission (DSE) properties (Supplementary Figs. 33–35).

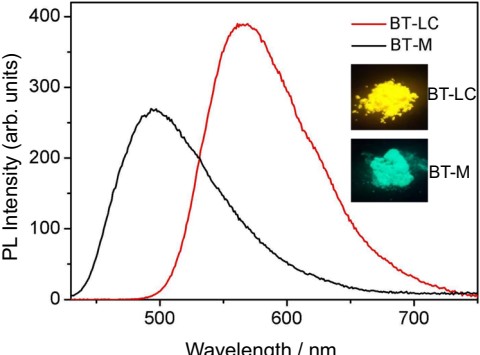

**Fig. 2** Photoluminescence spectra of BT-M and BT-LC in the solid state (insets: photographs in solid state under 365 nm UV illuminations).

**Fig. 1** The synthetic route of BT-LC.

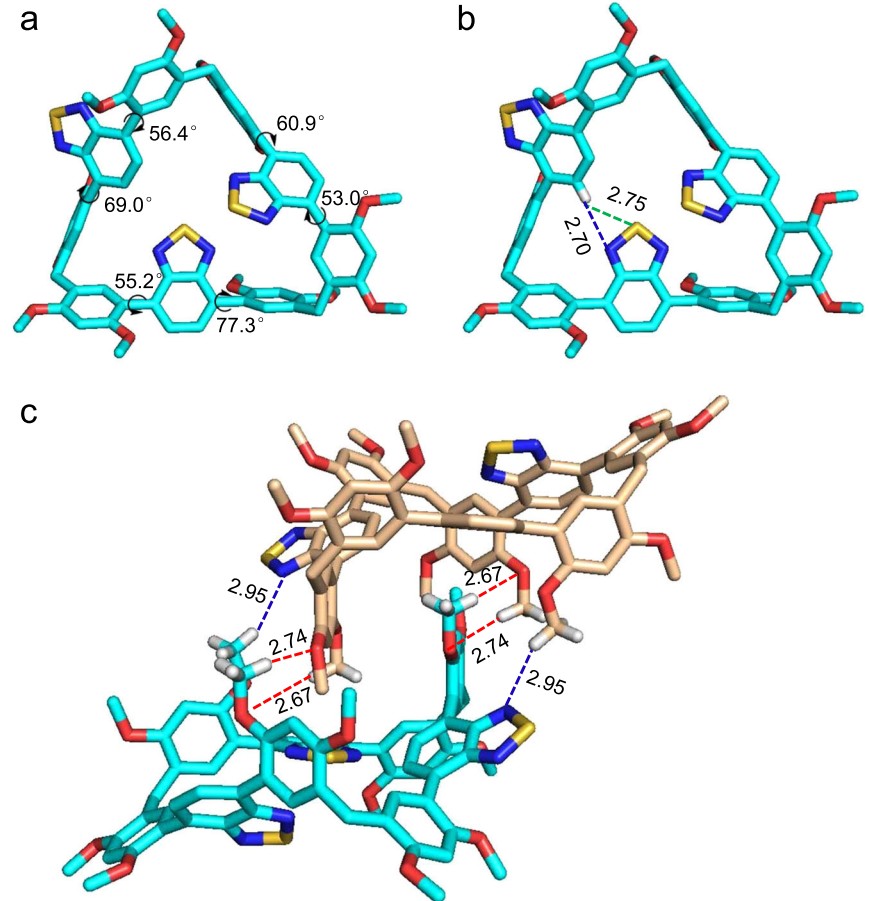

**Fig. 3 Single-crystal X-ray diffraction analysis of BT-LC. a** Torsion angles of BT-LC. **b** Intramolecular interactions of BT-LC: C–H⋯N (blue lines), C–H⋯S (green lines). **c** Intermolecular interactions of BT-LC: C–H⋯O (red lines); C–H⋯N (blue lines). For clarity, some hydrogen atoms and solvents are omitted.

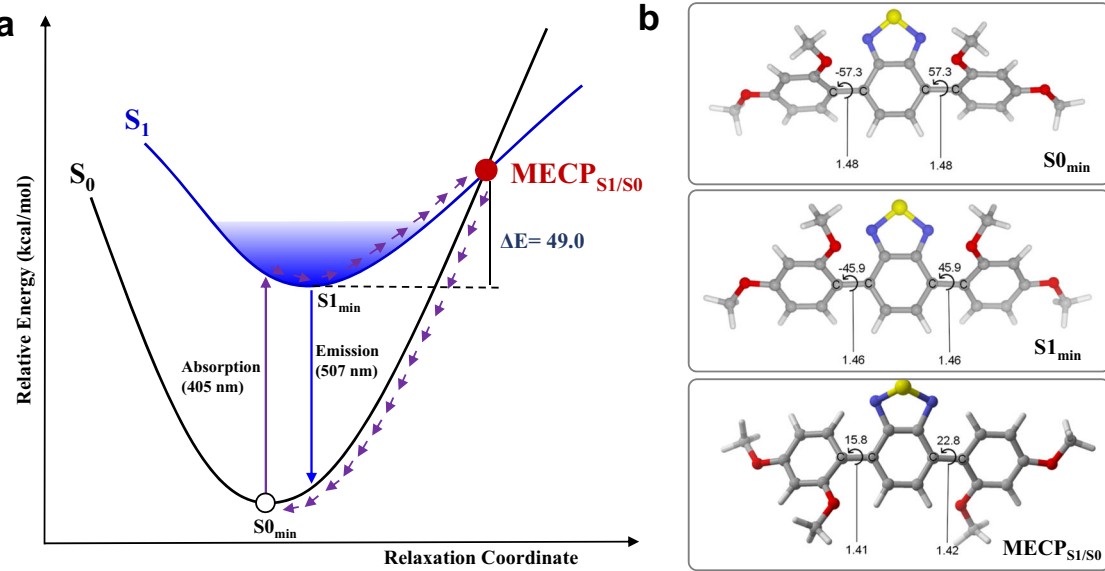

**Fig. 4 Radiative and non-radiative relaxation process of BT-M calculated at TDA-PBE0/PBE0/6-31G\* level. a** The non-radiative relaxation process of BT-M via minimum energy crossing point (MECP$_{S1/S0}$); **b** the minimum energy structures of BT-M in ground state (S0$_{min}$), singlet state (S1$_{min}$), and MECP$_{S1/S0}$. The selected bond lengths are in Å and the selected torsion angles are in degree.

Single crystal analysis was examined to understand macrocyclic effects. From Fig. 3a and Supplementary Fig. 48, BT-LC has a rigid triangular geometry with vertex angles of 112°, 114°, and 115°. The $C(sp^3)$ bridges spatially separated three luminophores in a single macrocyclic molecule, leading to the alleviation of the concentration quenching to a certain degree. Three series of torsion angles between the central benzothiadiazole planes and their adjacent phenyl rings are 55.29°, 77.35°; 53.02°, 60.92°; and 56.40°, 69.05, respectively. In comparison, both torsion angles for the monomer BT-M are 63.70° (Supplementary Fig. 45a). Macrocyclization of luminophores would efficiently reduce the space of their rotation, therefore preventing their rotation. Furthermore, the intramolecular hydrogen bonds (C–H···N 2.70 Å, and C–H···S 2.75 Å, Fig. 3b) between benzothiadiazoles could further limit the intramolecular motion. It should be noted that BT-LC are racemes possessing a pair of enantiomers with double-included dimer geometry, where dimethoxyphenyl group of one enantiomer was encapsulated in the cavity of the other one through multiple intermolecular interactions (C–H···O 2.74 Å, 2.67 Å, and C–H···N 2.95 Å, Fig. 3c). For the monomer, adjacent benzothiadiazole units were packed in an antiparallel stacking with a separation of about 4.98 Å (Supplementary Fig. 45b); no C–H···N/S hydrogen bonds were found. Obviously, the tight packing arrangement of BT-LC is more beneficial to fluorescence than that of BT-M. Thus, the above results suggest that MIEE effect should be due to the alleviation of quenching caused by spatial separation and the restriction of intramolecular motion by macrocyclic topologic structure and intra/intermolecular interactions.

**Mechanism study of MIEE.** To further illustrate the principle of MIEE, we calculated the process of radiative relaxation and non-radiative relaxation. As a competition of luminescence process, non-radiative relaxation process is semi-quantitatively described using the TDA-PBE0 method[49,50]. As shown in the Fig. 4a, BT-M can return to the ground state through a $MECP_{S1/S0}$, resulting in the decrease of fluorescence efficiency. It is particularly noteworthy that the C–C bond between benzothiadiazole and adjacent phenyl rings is gradually shortened according to the order of $S0_{min}$, $S1_{min}$, and $MECP_{S1/S0}$. Among them, the C–C bond length in $MECP_{S1/S0}$ is only 1.41 Å (Fig. 4b), which is distinctly shorter than the one of C–C single bond. With the double-bonding tendency, the non-radiative relaxation process requires that the torsion angle between benzothiadiazole and benzene ring can be twisted to near 20° at $MECP_{S1/S0}$. That is to say, benzothiadiazole and benzene ring tend to be in the same plane at $MECP_{S1/S0}$. Unlike BT-M, BT-LC has a rigid triangular geometry and the rotation of the corresponding torsion angle would be limited (Fig. 3a and Supplementary Fig. 48). Therefore, BT-LC can avoid the process of $MECP_{S1/S0}$ non-radiative relaxation and its fluorescence efficiency is enhanced. These calculation results are consistent with our assumption at the beginning of the article. It should be pointed out that the method remains computationally too expensive to apply to large BT-LC systems.

**Electroluminescent (EL) properties.** We then explore the possibility of applying BT-LC as OLED emitters. Thermogravimetric analysis (TGA) was first performed to assess its thermal stability. From Supplementary Fig. 51, the macrocycle has a higher decomposition temperature [Td (5 wt% loss) 423–426 °C] than the monomer (286–291 °C), demonstrating rigidity can increase the decomposition temperature. On the other hand, the spectral stabilities of BT-M and BT-LC were examined. The pristine BT-M displayed green fluorescence peaking at 491 nm. After grinding for 2 min, its maximum emission wavelength red-shifts to 505 nm and emits yellow-green fluorescence (Supplementary Fig. 35d). For BT-

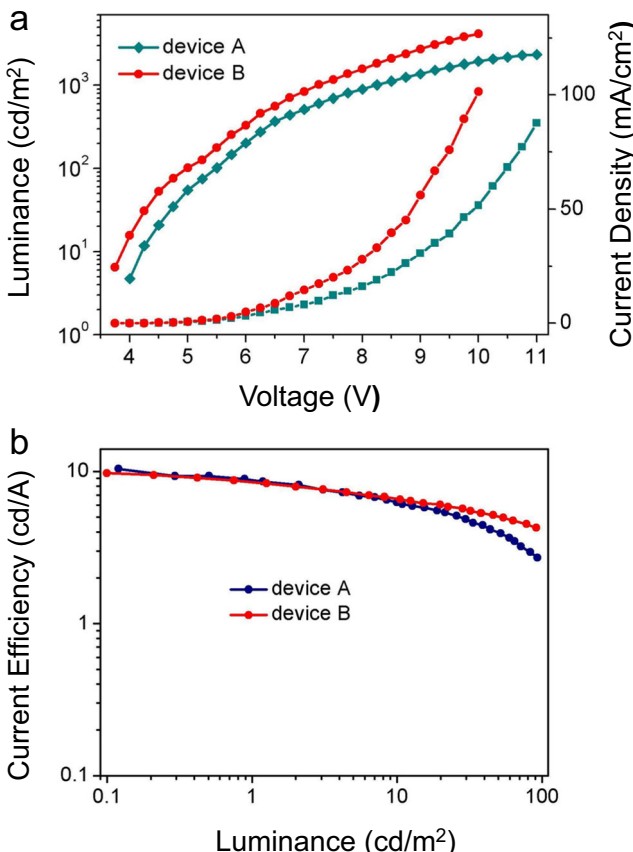

**Fig. 5 The performance of devices A and B. a** Current density–voltage–luminance (J–V–L) characteristics of A and B; **b** plots of current efficiency versus luminance.

LC (Supplementary Fig. 35c), the emission wavelength and color of the pristine and ground sample were hardly affected by external stimuli, implying that macrocyclization can improve spectral stability.

High thermal decomposition temperatures suggested that the coevaporation technique is very suitable for purification of OLEDs. Subsequently, OLEDs were fabricated with a multilayer configuration (Supplementary Fig. 52): ITO (100 nm)/ HATCN (10 nm)/TAPC (40 nm)/mCP: emitter (20 nm)/TmPyPB (40 nm)/Liq (2 nm)/Al (100 nm). The device A fabricated with monomer and the device B fabricated with macrocycle at 5% doping concentration were compared to reflect the performance of OLEDs. As shown in Fig. 5, Supplementary Figs. 53–55 and Table 1, device B exhibited higher $B_{max}$, $PE_{max}$, and $EQE_{max}$ than device A did. In other words, the macrocycle gives better device performance than the monomer which could be ascribed to improved quantum yield and rigidity caused by macrocyclization. Certainly, compared with reported BT-based emitters, macrocycle BT-LC showed moderate $CE_{max}$, $PE_{max}$, and $EQE_{max}$ (Supplementary Fig. 56 and Supplementary Table 4). Although the performance of the device is inferior to that of the current state-of-art ($EQE_{max}$, 8.47%), it is the first example of macrocyclic arene-based OLED. Macrocycles would be potentially applicated in OLEDs considering the following two points: (1) our modular synthesis method could conveniently produce diverse fluorescence macrocycles[34], (2) MIEE is an efficient strategy to improve $\Phi_{PL}$ values of chromophores.

**Application scope of MIEE.** To expand the application scope of MIEE, two kinds of macrocycles were designed and synthesized:

**Table 1 Summary of the EL data of devices A and B.**

| Device | Emitting layer | Dopant ratio (wt%) | $V_{on}$ (V)[a] | $B_{max}$ (cd/m$_2$)[b] | $CE_{max}$ (cd/A)[c] | $PE_{max}$ (lm/w$_2$)[d] | $EQE_{max}$ (%)[e] | $\lambda_{em}$ (nm)[f] | CIE (x, y)[g] |
|---|---|---|---|---|---|---|---|---|---|
| A | BT-M | 5 | 3.98 | 2369 | 10.10 | 7.10 | 1.92 | 512 | (0.29,0.53) |
| B | BT-LC | 5 | 3.82 | 4355 | 9.93 | 8.25 | 2.82 | 534 | (0.35,0.57) |

[a]Turn-on voltage.
[b]Maximum brightness ($B_{max}$).
[c]Maximum current efficiency ($CE_{max}$).
[d]Maximum power efficiency ($PE_{max}$).
[e]Maximum external quantum efficiency ($EQE_{max}$).
[f]EL peak wavelength.
[g]Commission International de l'Eclairage coordinates.

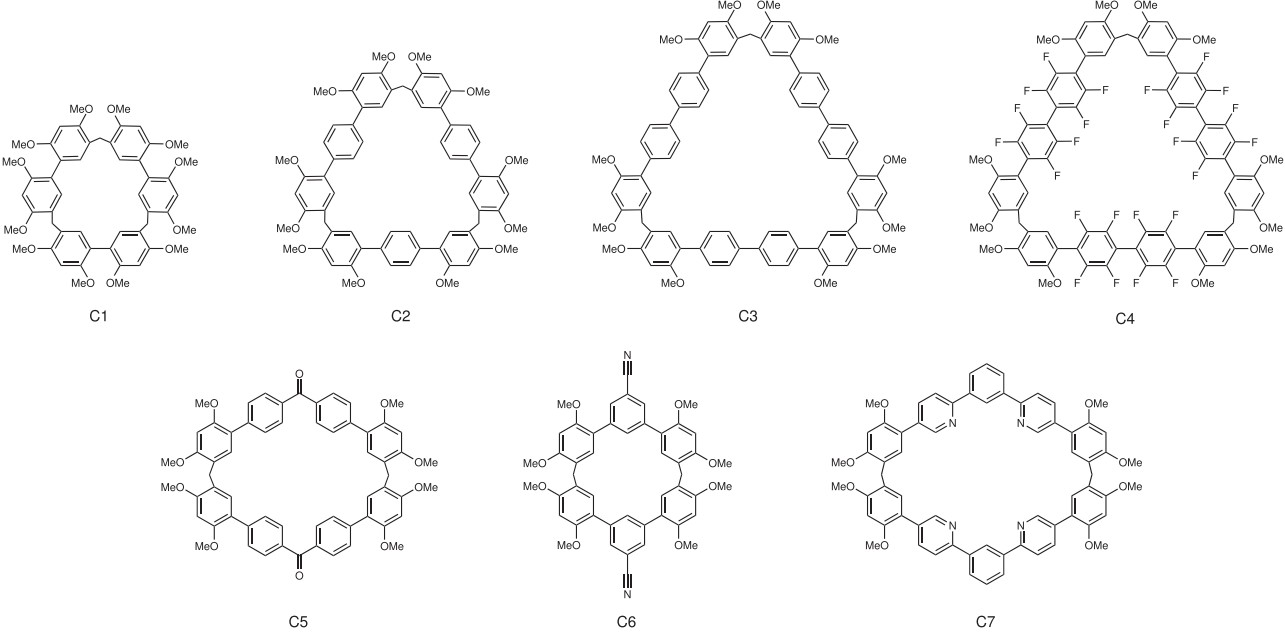

**Fig. 6** Chemical structure of other macrocycles C1–7.

**Table 2 Solid-state photophysical properties of monomers and corresponding macrocycles.**

|  | M1 | C1 | M2 | C2 | M3 | C3 | M4 | C4 | M5 | C5 | M6 | C6 | M7 | C7 |
|---|---|---|---|---|---|---|---|---|---|---|---|---|---|---|
| $\lambda_{ex}$(nm) | 300 | 308 | 300 | 332 | 325 | 332 | 310 | 320 | 350 | 405 | 340 | 300 | 300 | 310 |
| $\lambda_{em}$(nm) | 355 | 384 | 384 | 384 | 403 | 408 | 366 | 392 | 468 | 450 | 361 | 373 | 390 | 449 |
| $\Phi_{PL}$(%) | 5.51 | 19.2 | 15.3 | 53.0 | 34.4 | 37.5 | 12.7 | 65.2 | 0.6 | 5.9 | 12.1 | 17.5 | 21.9 | 28.2 |

$\lambda_{ex}$ (nm): excitation maximum.
$\lambda_{em}$ (nm): fluorescence maximum.
$\Phi_{PL}$: absolute PL quantum yield.

one is cyclic trimer with triangle shape (C1–C4), and the other is cyclic dimer with tetragonum shape (C5–C7). They were prepared according to our recently reported method, where linear monomer produces trimers, while V-shaped one tends to form dimer (Fig. 6 and Supplementary Figs. 1 and 8–31)[34]. Photophysical properties of these macrocycles and monomers were shown in Supplementary Figs. 38–44 and summarized in Table 2. Similar to BT-LC, the other triangular macrocycles (C1–C4) also showed MIEE, especially for octafluorobenzene-based C4, whose $\Phi_{PL}$ value greatly increased from 12.7 to 65.2%. For tetragonal macrocycles C5–C7, their quantum yields were also improved compared with corresponding monomers. In particular, the $\Phi_{PL}$ value of C5 is 9.8 times than that of M5. The results showed that MIEE is an effective and general strategy to enhance solid-state

emitters, although for macrocycles C3 and C7, emission improvement is not significant.

## Discussion

In summary, we have developed an effective and universal strategy named "MIEE" to improve luminous efficiency of chromophores by linking chromophores into macrocycle through C(sp$^3$) bridges. Macrocycle BT-LC exhibited high fluorescence quantum yield, up to 99%, much higher than that of the monomer. Furthermore, the macrocycle BT-LC exhibited higher maximum brightness, power efficiency, and external quantum efficiency than the monomer in the application of OLEDs. Mechanism study revealed that the monomer can return to the

ground state through a minimum energy crossing point ($MECP_{S1/S0}$) which decreases the fluorescence efficiency. After macrocyclization, the inherent structural rigidity prohibits this non-radiative relaxation process and the radiative relaxation is dominant, therefore showing intense emission. Moreover, MIEE strategy has good universality that several macrocycles with different luminophores work well. MIEE is a new and effective approach to improve the luminophore's emission and would be helpful to developing organic luminescent materials.

## Methods

**General**. All reagents were purchased commercially and used without further purification unless otherwise noted. $^1H$ NMR and $^{13}C$ NMR spectra were recorded on Bruker Avance III 400 MHz, Bruker Avance III 500 MHz and Bruker Avance III 600 MHz. HRMS was determined on Bruker Daltonics AutoflexIII LRF200-CID, Bruker Daltonics Inc. APEXIII 7.0 TESLA FTMS and Agilent 6520 q-TOF LC-MS. UV–vis spectra were taken on a UV-2501PC UV–vis recording spectrophotometer (Shimadzu). Quantum efficiency was measured on HAMAMATSU C9920-02. Melting points were obtained on an X-4 digital melting point apparatus without correction. Single crystal X-ray diffraction data of BT-M were collected on a Bruker APXE II CCD detector using Mo-Kα radiation (λ = 0.71074 Å). Single crystal X-ray diffraction data of BT-LC were determined on Bruker D8 Venture using Mo-Kα radiation (λ = 0.71073 Å). TGA was recorded using a TA Instrument TA-Q500 and the samples were heated under nitrogen gas at a rate of 10 °C/min. The current density–voltage–luminance characteristics, electroluminescent (EL) spectra, $CIE_{x,y}$ coordinates of devices are measured and recorded by computer-controlled PR655 spectrometer and Keithley 2400 digital power. All the calculations of ground states were performed at the PBE0/6-31g* level[51–53] using the Gaussian16 suite of programs[54]. For excited state calculation, The Tamm–Dancoff approximation (TDA)[49] was used for TDDFT because it is more stable near MECP[50]. Harmonic vibration frequency calculations were used to confirm the stationary points. $MECP_{S1/S0}$ is located at the TDA-PBE0/PBE0/6-31G* level using the Newton-Lagrange method, which was introduced by Koga and Morokuma[55]. These calculations were treated using a homemade program LookForMECP (version 2.1). This program can be obtained from the authors upon request. The early version of this program had been used successfully to search the MECP[56–60]. The 3D figures of molecular structure were prepared by CYLView[61].

**Synthesis and characterization**. Synthesis and relevant characterization details are provided in the Supplementary Information.

## Data availability

The authors declare that the data supporting the findings of this study are available within the paper and its Supplementary Information. All data are available from the authors on request. Crystallographic data for the structures reported in this Article have been deposited at the Cambridge Crystallographic Data Center, under deposition numbers CCDC 2074796 (BT-M) and 2074805 (BT-LC). Copies of the data can be obtained free of charge via https://www.ccdc.cam.ac.uk/structures/.

## Code availability

The computational program used for locating to MECP in this manuscript is available in GitHub[62] (https://github.com/bnulk/LookForMECP).

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

## Acknowledgements

Financial support from the National Natural Science Foundation of China (21971192, 21772118) (L.C.) (21571126) (B.Y.L.) and the Natural Science Foundation of Tianjin City (20JCZDJC00200) (L.C.) are gratefully acknowledged.

## Author contributions

L.S., C.L., B.Y.L., and L.C. conceived this project and designed the experiments; L.S. contributed to most of the experimental work; L.K. and L.M. contributed theoretical calculations; F.X.-C., L.Z.-X., and W.B. synthesized partial macrocycles; Z.Z.-Y. designed and analyzed the experiments.

## Competing interests

The authors declare no competing interests.
