## [Peer review file · Nature Communications]

REVIEWER COMMENTS

Reviewer #1 (Remarks to the Author):

In this manuscript, Li et al reported a benzothiadiazole-derived macrocycle. The macrocycle was prepared by well-developed chemistry through condensation of electron-rich arenes with paraformaldehyde. The authors also claimed a concept of macrocyclization-induced emission enhancement (MIEE) because this newly synthesized macrocycle was found more fluorescent than its precursor. Overall, the manuscript reflects a reasonable amount of solid lab work. However, considering the high standards of Nat. Comm., favorable recommendation cannot be made for this manuscript based on the following reasons.

1. The introduction of the concept “macrocyclization-induced emission enhancement (MIEE)” seems to be a major selling point of this work. However, the manuscript is not well-organized to explain the molecular design, scope and limitation, and mechanism of this concept.

What are the considerations of molecular designs to trigger the so-called MIEE? How many different fluorophores can this concept be applicable to? And photophysically why?

2. It is not rare for an organic compound to be strongly fluorescent both in solution and in solid state. What would be really interesting is the special photophysical mechanism behind the observations. Unfortunately, this manuscript did not provide in-depth mechanistic investigation.

It seems confusing that the authors characterize the MIEE of the macrocycle using the analogous experiments of AIE. Such experiments were employed to trigger the formation of aggregates, but not macrocyclizations.

3. The authors' efforts to construct OLED devices based on the macrocycle should be applauded. But how does the devices' performance compared to the current state-of-art?

Reviewer #2 (Remarks to the Author):

In this paper, the authors described the synthesis of a benzothiadiazole-based macrocycle (BT-LC) and the macrocyclization-induced emission enhancement (MIEE) in the solid-state. Compared with the monomer 4,7-bis(2,4-dimethoxyphenyl)-2,1,3-benzothiadiazole (BT-M), the macrocycle BT-LC produces much more intense fluorescence in the solid state ($\Phi_{\text{PL}}=99\%$) and exhibits better device performance in the application of OLEDs. The MIEE can be ascribed to the restriction of intramolecular motion and the alleviation of the concentration quenching by the macrocyclic topological structure. Although the increased emission of the macrocycle in solid state or its AIE property is very interesting, but the phenomenon is general. Additionally, the performance of the device based on the macrocycle is not pleasantly surprised. Thus, I'm not sure that the paper is enough novelty to publish in Nat. Commun. Apparently, one stunning application for the macrocycle with strong emission will be powerful to improve the quality of the manuscript.

Reviewer #3 (Remarks to the Author):

Organic luminescent materials have drawn much attention in the past decades. Some effective methods have been built to enhance the emission efficiency, for example aggregation induced emission and crystallization induced emission. In this manuscript, authors developed a novel strategy, namely macrocyclization-induced emission enhancement (MIEE), for the improvement of luminophore's solid-state emission. A benzothiadiazole-based macrocycle (BT-LC) was found to show much more intense fluorescence in the solid state ($\Phi_{\text{PL}} = 99\%$) and exhibit better device performance in the application of OLEDs, in comparison with the corresponding monomer. Moreover, BT-LC showed strong emission in both dilute solution and aggregated state. MIEE is a new and effective approach to improve the luminophore's emission and would be helpful to developing organic luminescent materials. Hence, I believe that this work is highly suitable for publication after considering the following minor issues.

1. In assessment of solid-state photophysical properties, please add fluorescence lifetime of the monomers and the macrocycles reported.
2. For the synthesis, are there any other cyclic oligomers such as tetramer as the by-product?
3. In page 6, "BT-BC" should be BT-LC.
4. Please define the abbreviated terms when they appear for the first time. "Tol" or "TOL"? Please be consistent.

Notice that six new authors were added during the revision. Liu, K. and Li, M. contributed density functional theory; Feng, X-C., Li, Z-X. and Wang, B. synthesized part of macrocycles; Zhang, Z-Y. designed and analyzed the experiments.

Reply to Reviewer #1

Comments: In this manuscript, Li et al reported a benzothiadiazole-derived macrocycle. The macrocycle was prepared by well-developed chemistry through condensation of electron-rich arenes with paraformaldehyde. The authors also claimed a concept of macrocyclization-induced emission enhancement (MIEE) because this newly synthesized macrocycle was found more fluorescent than its precursor. Overall, the manuscript reflects a reasonable amount of solid lab work. However, considering the high standards of Nat. Comm., favorable recommendation cannot be made for this manuscript based on the following reasons.

Reply: We greatly appreciate the reviewer's comments and advices. Herewith, we addressed the referee's comments as follows:

1. The introduction of the concept "macrocyclization-induced emission enhancement (MIEE)" seems to be a major selling point of this work. However, the manuscript is not well-organized to explain the molecular design, scope and limitation, and mechanism of this concept.

What are the considerations of molecular designs to trigger the so-called MIEE? How many different fluorophores can this concept be applicable to? And photophysically why?

Reply: 1.1 Molecular design

Molecular design is mainly considering the following two points: 1) macrocyclization can restrict intramolecular motion by locking chromophores into the skeleton of macrocycle to suppress non-radiation relaxation; 2) it spatially separates chromophores in a single macrocycle to eliminate the concentration quenching to a

certain degree. Please see the discussion in Paragraph 2 Page 2:

“Such macrocyclization-induced emission enhancement (MIEE) is theoretically feasible considering the following two features: on one hand, it spatially separates chromophores in a single macrocycle to eliminate the concentration quenching to a certain degree,³⁵⁻³⁸ on the other hand, it restricts intramolecular motion by locking its chromophores into the skeleton of macrocycle to suppress non-radiation relaxation.^{39,40}”

1.2 Scope and limitation

To expand the application scope of MIEE, two kinds of macrocycles were designed and synthesized: one is cyclic trimer with triangle shape (C1-C4), and the other is cyclic dimer with tetragonum shape (C5-C7) (Figures R1).^{1,2} Photophysical properties of these macrocycles and monomers were shown in Figure R2-8 and summarized in Table R1. The results showed that MIEE is an effective and general strategy to enhance solid-state emitters. Figure R1 and Table R1 were added in the revised manuscript as Figure 6 and Table 2, respectively; Figure R2-8 were added in the revised Supplementary Information as Supplementary Figure S38-44; Figure R9 was added in the revised Supplementary Information as Supplementary Figure S1. Figure R10-33 were added in the revised Supplementary Information as Supplementary Figure S8-31.

The following discussion was added to the revised manuscript in Page 9:

*“**Application scope of MIEE.** To expand the application scope of MIEE, two kinds of macrocycles were designed and synthesized: one is cyclic trimer with triangle shape (C1-C4), and the other is cyclic dimer with tetragonum shape (C5-C7). They were prepared according to our recently reported method, where linear monomer produces trimers, while V-shaped one tends to form dimer (Figures 6 and S1).³⁴ Photophysical properties of these macrocycles and monomers were shown in Figure S38-44 and summarized in Table 2. Similar to BT-LC, the other triangular macrocycles (C1-C4) also showed macrocyclization-induced emission enhancement,*

especially for octafluorobenzene-based C4, whose Φ_{PL} value greatly increased from 12.7% to 65.2%. For tetragonal macrocycles C5-C7, their quantum yields were also improved compared with corresponding monomers. In particular, the Φ_{PL} value of C5 is 9.8 times than that of M5. The results showed that MIEE is an effective and general strategy to enhance solid-state emitters, although for macrocycles C3 and C7, emission improvement is not significant.”

Figure R1 (Figure 6). Chemical structure of other selected macrocycles.

Table R1 (Table 2). Solid-state photophysical properties of monomers and corresponding macrocycles

	M1	C1	M2	C2	M3	C3	M4	C4	M5	C5	M6	C6	M7	C7
$\lambda_{ex}(nm)$	300	308	300	332	325	332	310	320	350	405	340	300	300	310
$\lambda_{em}(nm)$	355	384	384	384	403	408	366	392	468	450	361	373	390	449
$\Phi_{PL}(\%)$	5.51	19.2	15.3	53.0	34.4	37.5	12.7	65.2	0.6	5.9	12.1	17.5	21.9	28.2

λ_{ex} (nm): excitation maximum. λ_{em} (nm): fluorescence maximum. Φ_{PL} : absolute PL quantum yield.

Figure R2 (Figure S38) PL spectra of C1 and M1 (Insets: photographs in solid state under 365 nm UV illuminations).

Figure R3 (Figure S39) PL spectra of C2 and M2 (Insets: photographs in solid state under 365 nm UV illuminations).

Figure R4 (Figure S40) PL spectra of C3 and M3 (Insets: photographs in solid state under 365 nm UV illuminations).

Figure R5 (Figure S41) PL spectra of C4 and M4 (Insets: photographs in solid state under 365 nm UV illuminations).

Figure R6 (Figure S42) PL spectra of C5 and M5 (Insets: photographs in solid state under 365 nm UV illuminations).

Figure R7 (Figure S43) PL spectra of C6 and M6 (Insets: photographs in solid state under 365 nm UV illuminations).

Figure R8 (Figure S44) PL spectra of C7 and M7 (Insets: photographs in solid state under 365 nm UV illuminations).

Synthesis characterization of new compounds

Figure R9 (Figure S1) Chemical structure of other monomers and corresponding macrocycles.

M4. Under the protection of N_2 atmosphere, 4,4'-dibromooctafluorobiphenyl (4.56 g, 10.0 mmol), 2,4-dimethoxyphenylboronic acid (4.55 g, 25.0 mmol), and tetrakis(triphenylphosphine)palladium(0) (1.16 g, 1.00 mmol) were dissolved in tetrahydrofuran (160 mL). The sodium carbonate (4.24 g, 40.0 mmol) in water (25 mL) was added into the solution and stirred for 24 h at 85 °C. Upon cooling to room temperature, water (120 mL), dichloromethane (120 mL) was added and stirred. After filtration of the solution, the solution was partitioned between dichloromethane and water. The product was extracted from the organic layer. The aqueous layer was further extracted twice with dichloromethane (120 mL). The combined organic layer was dried over anhydrous Na_2SO_4 and evaporated under reduced pressure. The product was purified by column chromatography on silica gel (eluent: 4/1, v/v, dichloromethane : petroleum ether) to give product M4 (4.33g, 76%) as a white solid. m.p. 198-199 °C; 1H NMR (400 MHz, $CDCl_3$, 298 K) δ 7.25 (d, J = 8.0 Hz, 2H), 6.66-6.63 (m, 4H), 3.89 (s, 6H), 3.85 (s, 6H). ^{13}C NMR (100 MHz, $CDCl_3$, 298 K) δ 162.5, 158.5, 145.5, 143.5, 132.4, 119.7, 108.5, 106.1, 105.1, 99.1, 55.9, 55.6. HRMS (ESI) m/z: $[M+H]^+$ calcd for $[C_{28}H_{19}F_8O_4]^+$, 571.1150; found, 571.1156.

C4. To the solution of M4 (2.85g, 5.00 mmol) in dichloromethane (200 mL) was added paraformaldehyde (0.450 g, 15.0 mmol). Boron trifluoride diethyl etherate (0.650 ml, 5.00 mmol) was then added to the reaction mixture. The mixture was stirred at 25 °C for 30 minutes. Then the reaction was quenched by addition of 200 mL saturated aqueous $NaHCO_3$. The solution was partitioned between dichloromethane and saturated aqueous $NaHCO_3$. The product was extracted from the

organic layer. The aqueous layer was further extracted twice with dichloromethane (120 mL). The combined organic layer was dried over anhydrous Na_2SO_4 and concentrated. The product was purified by column chromatography on silica gel (eluent : dichloromethane: ethyl acetate = 3:1) to obtain product C4 (1.05 g, 36%) as a white solid. m.p. $>320^\circ\text{C}$; ^1H NMR (400 MHz, DMSO, 353 K) δ 8.39-6.68 (m, 12H), 3.92 (s, 18H), 3.86 (s, 18H), 3.81 (s, 6H). ^{13}C NMR (100 MHz, CDCl_3 , 298 K) δ 159.8, 156.9, 145.6, 143.1, 132.5, 131.0, 128.9, 121.4, 119.7, 107.4, 105.9, 95.7, 56.0, 55.8, 28.1. HRMS (ESI) m/z: $[\text{M}+\text{H}]^+$ calcd for $[\text{C}_{87}\text{H}_{55}\text{F}_{24}\text{O}_{12}]^+$, 1747.3305; found, 1747.3251.

M5. Under the protection of N_2 atmosphere, 4,4'-dibromobenzophenone (6.80 g, 20.0 mmol), 2,4-dimethoxybenzeneboronic acid (9.10 g, 50.0 mmol), and tetrakis(triphenylphosphine)palladium(0) (2.32 g, 2.00 mmol) were dissolved in tetrahydrofuran (350 mL). The sodium carbonate (8.48 g, 80.0 mmol) in water (50 mL) was added into the solution and stirred for 24 h at 85°C . Upon cooling to room temperature, water (200 mL), dichloromethane (200 mL) was added and stirred. After filtration of the solution, the solution was partitioned between dichloromethane and water. The product was extracted from the organic layer. The aqueous layer was further extracted twice with dichloromethane (200 mL). The combined organic layer was dried over anhydrous Na_2SO_4 and evaporated under reduced pressure. The product was purified by column chromatography on silica gel (eluent: 4/1, v/v, dichloromethane : petroleum ether) to give product M5 (8.54g, 94%) as a white solid. m.p. $175\text{-}176^\circ\text{C}$; ^1H NMR (600 MHz, CDCl_3 , 298 K) δ 7.90 (d, $J = 6.00$ Hz, 4H),

7.65 (d, $J = 6.00$ Hz, 4H), 7.32 (d, $J = 6.00$ Hz, 2H), 6.62-6.59 (m, 4H), 3.87 (s, 6H), 3.84 (s, 6H). ^{13}C NMR (150 MHz, CDCl_3 , 298 K) δ 196.3, 161.0, 157.7, 142.8, 135.8, 131.5, 130.0, 129.3, 122.5, 105.0, 99.2, 55.7, 55.6. HRMS (ESI) m/z : $[\text{M}+\text{H}]^+$ calcd for $[\text{C}_{29}\text{H}_{27}\text{O}_5]^+$, 455.1853; found, 455.1860.

C5. To the solution of M5 (0.910 g, 2.00 mmol) in chloroform (100 mL) was added paraformaldehyde (0.180 g, 6.00 mmol). Boron trifluoride diethyl etherate (0.260 ml, 2.00 mmol) was then added to the reaction mixture. The mixture was stirred at 25 °C for 3 hours. Then the reaction was quenched by addition of 100 mL saturated aqueous NaHCO_3 . The solution was partitioned between dichloromethane and saturated aqueous NaHCO_3 . The product was extracted from the organic layer. The aqueous layer was further extracted twice with dichloromethane (100 mL). The combined organic layer was dried over anhydrous Na_2SO_4 and concentrated. The product was purified by column chromatography on silica gel (eluent : dichloromethane: ethyl acetate = 8:1) to obtain product C5 (0.056 g, 6%) as a white solid. m.p. $>320^\circ\text{C}$; ^1H NMR (600 MHz, CDCl_3 , 298 K) δ 7.81 (d, $J = 12.0$ Hz, 8H), 7.57 (d, $J = 6.00$ Hz, 8H), 7.03 (s, 4H), 6.57 (s, 4H), 3.94 (s, 4H), 3.90 (s, 12H), 3.84 (s, 12H). ^{13}C NMR (150 MHz, CDCl_3 , 298 K) δ 196.5, 158.4, 156.1, 142.9, 135.6, 131.9, 130.1, 129.3, 121.7, 121.3, 96.0, 56.1, 55.9, 27.3. HRMS (ESI) m/z : $[\text{M}+\text{H}]^+$ calcd for $[\text{C}_{60}\text{H}_{53}\text{O}_{10}]^+$, 933.3633; found, 933.3648.

M6. Under the protection of N_2 atmosphere, 3,5-dibromobenzonitrile (5.22 g, 20.0 mmol), 2,4-dimethoxybenzeneboronic acid (9.10 g, 50.0 mmol), and tetrakis(triphenylphosphine)palladium(0) (2.32 g, 2.00 mmol) were dissolved in

tetrahydrofuran (300 mL). The sodium carbonate (6.36 g, 60.0 mmol) in water (40 mL) was added into the solution and stirred for 24 h at 85 °C. Upon cooling to room temperature, water (150 mL), dichloromethane (150 mL) was added and stirred. After filtration of the solution, the solution was partitioned between dichloromethane and water. The product was extracted from the organic layer. The aqueous layer was further extracted twice with dichloromethane (150 mL). The combined organic layer was dried over anhydrous Na₂SO₄ and evaporated under reduced pressure. The product was purified by column chromatography on silica gel (eluent: 4/1, v/v, dichloromethane : petroleum ether) to give product M5 (6.83 g, 91%) as a white solid. m.p. 185-186 °C; ¹H NMR (400 MHz, CDCl₃, 298 K) δ 7.82 (s, 1H), 7.74 (s, 2H), 7.26 (d, *J* = 8.0 Hz, 2H), 6.60-6.58 (m, 4H), 3.87 (s, 6H), 3.83 (s, 6H). ¹³C NMR (100 MHz, CDCl₃, 298 K) δ 161.1, 157.5, 139.3, 134.9, 131.3, 131.1, 121.4, 119.7, 111.7, 105.0, 99.0, 55.7, 55.6. HRMS (ESI) *m/z*: [M+H]⁺ calcd for [C₂₃H₂₂NO₄]⁺, 376.1543; found, 376.1540.

C6. To the solution of M6 (1.86 g, 5.0 mmol) in 1,2-dichloroethane (150 mL) was added paraformaldehyde (0.450 g, 15.00 mmol). Boron trifluoride diethyl etherate (0.650 ml, 5.00 mmol) was then added to the reaction mixture. The mixture was stirred at 25 °C for 45 minutes. Then the reaction was quenched by addition of 150 mL saturated aqueous NaHCO₃. The solution was partitioned between dichloromethane and saturated aqueous NaHCO₃. The product was extracted from the organic layer. The aqueous layer was further extracted twice with dichloromethane (150 mL). The combined organic layer was dried over anhydrous Na₂SO₄ and concentrated. The product was purified by column chromatography on silica gel (eluent : dichloromethane: ethyl acetate = 20:1) to obtain product C6 (0.213 g, 11%) as a white solid. m.p. >320°C; ¹H NMR (400 MHz, CDCl₃, 298 K) δ 7.83 (s, 4H), 7.26 (s, 2H), 6.77 (s, 4H), 6.56 (s, 4H), 3.88 (s, 12H), 3.86 (s, 12H), 3.82 (s, 4H). ¹³C NMR (100 MHz, CDCl₃, 298 K) δ 158.6, 155.7, 139.9, 134.0, 131.5, 131.4, 121.0, 119.9, 110.9, 95.3, 55.8, 55.7, 28.5. HRMS (ESI) *m/z*: [M+Na]⁺ calcd for [C₄₈H₄₂N₂O₈]⁺, 797.2833; found, 797.2839.

M7. Under the protection of N₂ atmosphere, 1,3-bis(5-bromopyridin-2-yl)benzene (3.88 g, 10.0 mmol), 2,4-dimethoxybenzeneboronic acid (5.46 g, 30.0 mmol), and tetrakis(triphenylphosphine)palladium(0) (1.16 g, 1.00 mmol) were dissolved in tetrahydrofuran (200 mL). The sodium carbonate (3.18 g, 30.0 mmol) in water (30 mL) was added into the solution and stirred for 24 h at 85 °C. Upon cooling to room temperature, water (120 mL), dichloromethane (120 mL) was added and stirred. After filtration of the solution, the solution was partitioned between dichloromethane and water. The product was extracted from the organic layer. The aqueous layer was further extracted twice with dichloromethane (120 mL). The combined organic layer was dried over anhydrous Na₂SO₄ and evaporated under reduced pressure. The product was purified by column chromatography on silica gel (eluent: 5/1, v/v, dichloromethane : ethyl acetate) to give product M5 (4.48 g, 89%) as a pale yellow solid. m.p. 213-214; ¹H NMR (400 MHz, CDCl₃, 298 K) δ 8.87 (s, 2H), 8.70 (s, 1H), 8.11 (d, *J* = 8.0 Hz, 2H), 7.95-7.87 (m, 4H), 7.60 (s, 1H), 7.33 (d, *J* = 12.0 Hz, 2H), 6.64-6.60 (m, 4H), 3.87 (s, 6H), 3.84 (s, 6H). ¹³C NMR (100 MHz, CDCl₃, 298 K) δ 161.4, 158.2, 155.3, 150.4, 140.2, 137.7, 133.0, 131.4, 129.6, 127.6, 125.7, 120.3, 120.1, 105.4, 99.5, 56.0, 55.9. HRMS (ESI) *m/z*: [M+H]⁺ calcd for [C₃₂H₂₉N₂O₄]⁺, 505.2122; found, 505.2130.

C7. To the solution of M7 (1.08 g, 2.0 mmol) in 1,2-dichloroethane (200 mL) was added paraformaldehyde (0.180 g, 6.00 mmol). Boron trifluoride diethyl etherate (0.260 ml, 2.00 mmol) was then added to the reaction mixture. The mixture was stirred at 25 °C for 5 hours. Then the reaction was quenched by addition of 150 mL

saturated aqueous NaHCO₃. The solution was partitioned between dichloromethane and saturated aqueous NaHCO₃. The product was extracted from the organic layer. The aqueous layer was further extracted twice with dichloromethane (150 mL). The combined organic layer was dried over anhydrous Na₂SO₄ and concentrated. The product was purified by column chromatography on silica gel (eluent : dichloromethane: ethyl acetate = 1:1) to obtain product C7 (0.248 g, 24%) as a white solid. m.p. >320°C; ¹H NMR (400 MHz, CDCl₃, 298 K) δ 8.81 (s, 4H), 8.66 (s, 2H), 8.07 (d, *J* = 8.0 Hz, 4H), 7.90-7.86 (m, 8H), 7.58 (s, 2H), 6.99 (s, 4H), 6.59 (s, 4H), 3.93 (s, 4H), 3.90 (s, 12H), 3.84 (s, 12H). ¹³C NMR (100 MHz, CDCl₃, 298 K) δ 158.6, 156.3, 154.9, 150.2, 140.0, 137.5, 133.0, 131.7, 129.2, 127.1, 125.7, 121.8, 120.2, 120.0, 119.1, 105.3, 99.3, 96.1, 56.0, 55.4, 27.6. HRMS (ESI) *m/z*: [M+H]⁺ calcd for [C₆₆H₅₇N₄O₈]⁺, 1033.4171; found, 1033.4171.

Figure R10 (Figure S8) ¹H NMR spectrum (400 MHz, CDCl₃, 298 K) of M4.

Figure R11 (Figure S9) ^{13}C NMR spectrum (100 MHz, CDCl_3 , 298 K) of M4.

Figure R12 (Figure S10). HMRS spectrum of M4.

Figure R13 (Figure S11). ¹H NMR spectrum (400 MHz, DMSO, 353K) of C4.

Due to the isomerization, the protons of C4 are complicated even at high temperature of 353K. It was confirmed by the HRMS (**Figure R15**). We also got its X-ray crystal structure.

Figure R14 (Figure S12) ¹³C NMR spectrum (100 MHz, CDCl₃, 298 K) of C4.

Figure R15 (Figure S13) HMRS spectrum of C4.

Figure R16 (Figure S14) ¹H NMR spectrum (600 MHz, CDCl₃, 298 K) of M5.

Figure R17 (Figure S15) ¹³C NMR spectrum (150 MHz, CDCl₃, 298 K) of M5.

Figure R18 (Figure S16) HMRS spectrum of M5.

Figure R19 (Figure S17) ¹H NMR spectrum (600 MHz, CDCl₃, 298 K) of C5.

Figure R20 (Figure S18) ¹³C NMR spectrum (150 MHz, CDCl₃, 298 K) of M5.

Figure R21 (Figure S19) HMRS spectrum of C5.

Figure R22 (Figure S20) ^1H NMR spectrum (400 MHz, CDCl_3 , 298 K) of M6.

Figure R23 (Figure S21) ^{13}C NMR spectrum (100 MHz, CDCl_3 , 298 K) of M6.

Figure R24 (Figure S22) HMRS spectrum of M6.

Figure R25 (Figure S23) ¹H NMR spectrum (400 MHz, CDCl₃, 298 K) of C6 (* = petroleum ether peak signals).

Figure S26 (Figure S24) ^{13}C NMR spectrum (100 MHz, CDCl_3 , 298 K) of C6 (* = petroleum ether peak signals).

Figure R27 (Figure S25) HMRS spectrum of C6.

Figure R28 (Figure S26) ^1H NMR spectrum (400 MHz, CDCl_3 , 298 K) of M7 (* = ethyl acetate peak signals).

Figure R29 (Figure S27) ¹³C NMR spectrum (100 MHz, CDCl₃, 298 K) of M7 (* = ethyl acetate peak signals).

Figure R30 (Figure S28) HMRMS spectrum of M7.

Figure R31 (Figure S29) ¹H NMR spectrum (400 MHz, CDCl₃, 298 K) of C7 (* = ethyl acetate peak signals).

Figure R32 (Figure S30) ^{13}C NMR spectrum (100 MHz, CDCl_3 , 298 K) of C7 (* = ethyl acetate peak signals).

Figure R33 (Figure S31) HMRS spectrum of C7.

1.3 Mechanism of this concept

To illustrate the principle of MIEE, we calculated the process of radiative relaxation and non-radiative relaxation. As a competition of luminescence process, nonradiative relaxation process is semi-quantitatively described using the TDA-PBE0 method. As shown in the Figure R34a, BT-M can return to the ground state through a $\text{MECP}_{\text{S}_1/\text{S}_0}$, resulting in the decrease of fluorescence efficiency. It is particularly noteworthy that the C-C bond between benzothiadiazole and adjacent phenyl rings is gradually shortened according to the order of $\text{S}_{0\text{min}}$, $\text{S}_{1\text{min}}$ and $\text{MECP}_{\text{S}_1/\text{S}_0}$. Among them, the C-C bond length in $\text{MECP}_{\text{S}_1/\text{S}_0}$ is only 1.41 Å (Figure R34b), which is distinctly shorter than the one of C-C single bond. With the double-bonding tendency, the non-radiative relaxation process requires that the torsion angle between benzothiadiazole and benzene ring can be twisted to near 20 degrees at $\text{MECP}_{\text{S}_1/\text{S}_0}$. That is to say, benzothiadiazole and benzene ring tend to be in the same plane at $\text{MECP}_{\text{S}_1/\text{S}_0}$. Unlike BT-M, BT-LC has a rigid triangular geometry and the rotation of

the corresponding torsion angle would be limited (Figure 3a and S48). Therefore, BT-LC can avoid the process of $\text{MECP}_{S1/S0}$ non-radiative relaxation and therefore its fluorescence efficiency is enhanced. These calculation results are consistent with our assumption at the beginning of the article. It should be pointed out that the method remains computationally too expensive to apply to large BT-LC systems. Figure R34 was added in the revised manuscript as Figure 4.

The following discussion was added to the revised manuscript in Page 6:

“Mechanism study of MIEE. To further illustrate the principle of MIEE, we calculated the process of radiative relaxation and non-radiative relaxation. As a competition of luminescence process, non-radiative relaxation process is semi-quantitatively described using the TDA-PBE0 method.⁴⁹⁻⁵⁰ As shown in the Figure 4a, BT-M can return to the ground state through a $\text{MECP}_{S1/S0}$, resulting in the decrease of fluorescence efficiency. It is particularly noteworthy that the C-C bond between benzothiadiazole and adjacent phenyl rings is gradually shortened according to the order of $S0_{\text{min}}$, $S1_{\text{min}}$ and $\text{MECP}_{S1/S0}$. Among them, the C-C bond length in $\text{MECP}_{S1/S0}$ is only 1.41 Å (Figure 4b), which is distinctly shorter than the one of C-C single bond. With the double-bonding tendency, the non-radiative relaxation process requires that the torsion angle between benzothiadiazole and benzene ring can be twisted to near 20 degrees at $\text{MECP}_{S1/S0}$. That is to say, benzothiadiazole and benzene ring tend to be in the same plane at $\text{MECP}_{S1/S0}$. Unlike BT-M, BT-LC has a rigid triangular geometry and the rotation of the corresponding torsion angle would be limited (Figure 3a and S48). Therefore, BT-LC can avoid the process of $\text{MECP}_{S1/S0}$ non-radiative relaxation and its fluorescence efficiency is enhanced. These calculation results are consistent with our assumption at the beginning of the article. It should be pointed out that the method remains computationally too expensive to apply to large BT-LC systems.”

Figure R34 (Figure 4) Radiative and non-radiative relaxation process of BT-M calculated at TDA-PBE0/PBE0/6-31G* level. **a** The non-radiative relaxation process of BT-M via minimum energy crossing point (MECP_{S1/S0}); **b** the minimum energy structures of BT-M in ground state (S0_{min}), singlet state (S1_{min}), and minimum energy crossing point (MECP_{S1/S0}). The selected bond lengths are in Å and the selected torsion angles are in degree.

Quantum chemical calculation methods

All the calculations of ground states were performed at the PBE0/6-31g* level³ using the Gaussian16 suite of programs.⁴ For excited state calculation, The Tamm-Dancoff approximation (TDA)⁵ is used for TDDFT because it is more stable near minimum energy crossing point (MECP).⁶ Harmonic vibration frequency calculations are used to confirm the stationary points. MECP_{S1/S0} is located at the TDA-PBE0/PBE0/6-31G* level using the Newton-Lagrange method, which was introduced by Koga and Morokuma.⁷ These calculations are treated using a homemade program LookForMECP (version 2.1). This program can be obtained from the authors upon request. The early version of this program has been used successfully to search the MECP.⁸ The 3D figures of molecular structure were prepared by CYLView.⁹

Coordinates (Å) and energies (Hartree)

S0_{min}

E = -1657.336726 hartree

C	0.62416600	-1.29102100	0.13084600
C	-0.71323000	-1.28214100	0.20256300
C	1.39082400	-0.06492000	0.36866700
C	-1.40919900	-0.00794700	0.21488500
N	-1.36483300	-2.59721700	0.10475600
N	1.22798300	-2.62410200	-0.04674200
C	0.67565300	1.16800000	0.62509200
C	2.79846900	-0.03149100	0.25108400
C	-0.66501200	1.19953200	0.51767500
C	-2.80061900	0.08126200	-0.04765500
S	-0.02073200	-3.31610100	0.80461600
H	1.21697800	2.07610500	0.83790400
C	3.44881900	-1.11635000	-0.43584200
C	3.67073400	1.06531900	0.66854100
H	-1.19538500	2.13039600	0.65709400
C	-3.61927300	-1.07960900	0.15523900
C	-3.49446800	1.29135300	-0.46523400
C	4.73678400	-1.03446200	-0.88981900
H	2.83104100	-2.01221500	-0.56714900
C	4.96344900	1.14495100	0.19862700

O	3.18121500	1.94252800	1.54982700
C	-4.98892000	-1.03095000	0.11140800
H	-3.05786900	-1.99865000	0.36724700
C	-4.87196200	1.33375500	-0.49557300
O	-2.74028400	2.32021700	-0.87209600
C	5.49402500	0.11940100	-0.60665000
H	5.16577200	-1.86579200	-1.43658400
H	5.62398600	1.96114500	0.46141400
C	4.01265100	2.98933300	2.01473200
C	-5.62573700	0.18547500	-0.19058200
H	-5.56150900	-1.92894400	0.31156100
H	-5.41660200	2.22126300	-0.79129300
C	-3.37640900	3.49641100	-1.33215400
O	6.74835300	0.31466000	-1.00323800
H	4.90246100	2.58835500	2.51331900
H	4.31419200	3.65234800	1.19534300
H	3.40941100	3.54473800	2.73286700
O	-6.94662600	0.35481700	-0.24799300
H	-4.00478800	3.28499500	-2.20520500
H	-3.98343500	3.95750500	-0.54399500
H	-2.57065600	4.17321500	-1.61725400
C	7.38520100	-0.67865700	-1.79692700
C	-7.79300100	-0.75846800	0.00129400
H	7.47975400	-1.61850000	-1.24405200
H	6.83547700	-0.84653100	-2.72829700
H	8.37558700	-0.28239900	-2.01982400
H	-7.62178300	-1.55315200	-0.73194000
H	-7.64153000	-1.14732700	1.01350200
H	-8.81037200	-0.38012600	-0.09791500

S1_{min}

E = -1657.235636 hartree

C	1.42952000	-0.60458200	0.23031300
C	0.69347800	-1.66353500	0.79348900
C	0.71473200	0.45474800	-0.41033000
C	-0.69331400	-1.66364200	0.79352600
H	1.22788500	-2.46228500	1.30045400
C	2.88463000	-0.66117100	0.23172800
C	-0.71492200	0.45464700	-0.41027800
N	1.28355800	1.44487000	-1.12789000
C	-1.42952400	-0.60477500	0.23041400
H	-1.22760000	-2.46248000	1.30048600
C	3.55286200	-1.85321500	-0.08548900
C	3.69649400	0.47944500	0.52599600
N	-1.28397100	1.44467700	-1.12777500
S	-0.00029400	2.32234000	-1.74064700
C	-2.88461200	-0.66143300	0.23187600
C	4.92961700	-1.92752000	-0.21119200
H	2.95281700	-2.73112500	-0.30627600
C	5.07615100	0.42854200	0.37256300
O	3.04811300	1.53798000	1.01199800
C	-3.55296100	-1.85343100	-0.08521600
C	-3.69638000	0.47933000	0.52604500

C	5.69477000	-0.76670500	0.00008100
H	5.39562500	-2.86283500	-0.49843200
H	5.70597300	1.28721100	0.57182100
C	3.70339500	2.78642800	1.03737000
C	-4.92971600	-1.92753800	-0.21099500
H	-2.95301600	-2.73144500	-0.30586400
C	-5.07602900	0.42863300	0.37244300
O	-3.04788300	1.53770100	1.01217800
O	7.03623100	-0.71415500	-0.11061500
H	4.51966700	2.79647200	1.77148100
H	4.09021900	3.04025800	0.04438400
H	2.94406100	3.51226500	1.32865300
C	-5.69475100	-0.76658100	0.00005200
H	-5.39584300	-2.86281600	-0.49816700
H	-5.70572100	1.28742700	0.57156700
C	-3.70277800	2.78634500	1.03727200
C	7.72968800	-1.89046900	-0.47065800
O	-7.03617600	-0.71391700	-0.11083100
H	-4.08935000	3.04017300	0.04418800
H	-4.51917000	2.79672300	1.77124900
H	-2.94327900	3.51200000	1.32857800
H	7.58291300	-2.68518200	0.27111900
H	7.41955800	-2.25105400	-1.45899300
H	8.78492200	-1.61603700	-0.50146000
C	-7.72977700	-1.89029400	-0.47040900
H	-7.41986000	-2.25117100	-1.45870400
H	-7.58290900	-2.68479800	0.27156800
H	-8.78499600	-1.61579300	-0.50108700

MECP_{S1/S0}
E = -1657.157554 hartree

C	0.62416600	-1.29102100	0.13084600
C	-0.71323000	-1.28214100	0.20256300
C	1.39082400	-0.06492000	0.36866700
C	-1.40919900	-0.00794700	0.21488500
N	-1.36483300	-2.59721700	0.10475600
N	1.22798300	-2.62410200	-0.04674200
C	0.67565300	1.16800000	0.62509200
C	2.79846900	-0.03149100	0.25108400
C	-0.66501200	1.19953200	0.51767500
C	-2.80061900	0.08126200	-0.04765500
S	-0.02073200	-3.31610100	0.80461600
H	1.21697800	2.07610500	0.83790400
C	3.44881900	-1.11635000	-0.43584200
C	3.67073400	1.06531900	0.66854100
H	-1.19538500	2.13039600	0.65709400
C	-3.61927300	-1.07960900	0.15523900
C	-3.49446800	1.29135300	-0.46523400
C	4.73678400	-1.03446200	-0.88981900
H	2.83104100	-2.01221500	-0.56714900
C	4.96344900	1.14495100	0.19862700
O	3.18121500	1.94252800	1.54982700
C	-4.98892000	-1.03095000	0.11140800

H	-3.05786900	-1.99865000	0.36724700
C	-4.87196200	1.33375500	-0.49557300
O	-2.74028400	2.32021700	-0.87209600
C	5.49402500	0.11940100	-0.60665000
H	5.16577200	-1.86579200	-1.43658400
H	5.62398600	1.96114500	0.46141400
C	4.01265100	2.98933300	2.01473200
C	-5.62573700	0.18547500	-0.19058200
H	-5.56150900	-1.92894400	0.31156100
H	-5.41660200	2.22126300	-0.79129300
C	-3.37640900	3.49641100	-1.33215400
O	6.74835300	0.31466000	-1.00323800
H	4.90246100	2.58835500	2.51331900
H	4.31419200	3.65234800	1.19534300
H	3.40941100	3.54473800	2.73286700
O	-6.94662600	0.35481700	-0.24799300
H	-4.00478800	3.28499500	-2.20520500
H	-3.98343500	3.95750500	-0.54399500
H	-2.57065600	4.17321500	-1.61725400
C	7.38520100	-0.67865700	-1.79692700
C	-7.79300100	-0.75846800	0.00129400
H	7.47975400	-1.61850000	-1.24405200
H	6.83547700	-0.84653100	-2.72829700
H	8.37558700	-0.28239900	-2.01982400
H	-7.62178300	-1.55315200	-0.73194000
H	-7.64153000	-1.14732700	1.01350200
H	-8.81037200	-0.38012600	-0.09791500

2. It is not rare for an organic compound to be strongly fluorescent both in solution and in solid state. What would be really interesting is the special photophysical mechanism behind the observations. Unfortunately, this manuscript did not provide in-depth mechanistic investigation.

It seems confusing that the authors characterize the MIEE of the macrocycle using the analogous experiments of AIE. Such experiments were employed to trigger the formation of aggregates, but not macrocyclizations.

Reply: This is an excellent advice from the reviewer. We agree the reviewer's opinion that characterizing the MIEE of the macrocycle by the analogous experiments of AIE is not suitable. Therefore, related discussion and characterization were moved to the Supplementary information. Please see Supplementary Figure S35 and corresponding discussion in the revised Supplementary Information.

As mentioned above, we carried out time-dependent DFT (TD-DFT) calculations to investigate the in-depth mechanism. Please refer to "**1.3 mechanism of this concept**" (Page 25 in this file).

3. The authors' efforts to construct OLED devices based on the macrocycle should be applauded. But how does the devices' performance compared to the current state-of-art?

Reply: Thanks for this great suggestion. We summarized the electroluminescent properties based on BT emitters in Table R2 and Figure R35. Among the BT-based OLEDs, the macrocycle (BT-LC) showed medium current efficiency (CE_{max} , 9.93 cd A^{-1}), power efficiency (PE_{max} , 8.25 lm W^{-1}) and external quantum efficiency (EQE_{max} , 2.82%). Figure R35 and Table R2 were added in the revised Supplementary Information as Supplementary Figure S56 and Table S4.

The following discussion was added to the revised manuscript in Page 8:

"Certainly, compared with reported BT-based emitters, macrocycle BT-LC showed moderate CE_{max} , PE_{max} and EQE_{max} (Figure S56 and Table S4). Although the performance of the device is inferior to that of the current state-of-art (EQE_{max} , 8.47%), it is the first example of macrocyclic arene-based OLED. Macrocycles would

be potentially applied in OLEDs considering the following two points: 1) our modular synthesis method could conveniently produce diverse fluorescence macrocycles;³⁴ 2) MIEE is an efficient strategy to improve Φ_{PL} values of chromophores.”

Figure R35 (Figure S56). Chemical structures of benzothiadiazole-based emitters.

Table R2 (Table S4) Electroluminescence properties of benzothiadiazole-based emitters

	CE_{\max} (cd A^{-1})	PE_{\max} (lm W^{-1})	EQE_{\max} (%)	Ref.
BT-M	10.1	7.10	1.92	This work
BT-LC	9.93	8.25	2.82	This work
1	0.28	0.15	0.15	10
2	0.68	0.51	0.32	10
3	0.88	0.64	0.40	10
4	1.37	- ^a	1.00	11
5	0.4	0.5	1.00	12
6	5.2	3.0	1.50	12
7	6.4	2.9	3.10	12
8	- ^a	- ^a	1.43	13
9	- ^a	- ^a	1.73	13
10	1.31	1.59	2.17	14
11	1.41	1.70	2.03	14
12	2.19	1.61	2.09	14
13	6.25	5.17	- ^a	15
14	6.5	2.6	2.39	16
15	6.2	11.6	4.5	17
16	15.7	12.2	4.6	17
17	15.2	10.9	4.8	17
18	30.4	23.67	8.47	18

^a means that this value is not given in the original reference.

Reply to Reviewer #2

Comments: In this paper, the authors described the synthesis of a benzothiadiazole-based macrocycle (BT-LC) and the macrocyclization-induced emission enhancement (MIEE) in the solid-state. Compared with the monomer 4,7-bis(2,4-dimethoxyphenyl)-2,1,3-benzothiadiazole (BT-M), the macrocycle BT-LC produces much more intense fluorescence in the solid state ($\Phi_{\text{PL}}=99\%$) and exhibits better device performance in the application of OLEDs. The MIEE can be ascribed to the restriction of intramolecular motion and the alleviation of the concentration quenching by the macrocyclic topological structure. Although the increased emission of the macrocycle in solid state or its AIE property is very interesting, but the phenomenon is general. Additionally, the performance of the device based on the macrocycle is not pleasantly surprised. Thus, I'm not sure that the paper is enough novelty to publish in Nat. Commun. Apparently, one stunning application for the macrocycle with strong emission will be powerful to improve the quality of the manuscript.

Reply: We greatly appreciate the reviewer's advices. Herewith, we addressed the referee's comments as follows: Organic luminescent materials with high quantum efficiencies have attracted intensive attention. However, most of organic luminogens suffer from severe quenching effect in the aggregate state due to the formation of such detrimental aggregates as excimers and exciplexes, which greatly limits their applications in organic luminescent materials. We presented a novel strategy for the improvement of luminophore's solid-state emission, i.e., macrocyclization-induced emission enhancement (MIEE), by linking luminophores through $\text{C}(\text{sp}^3)$ bridges to give a macrocycle. We think this work has the novelty to be published in Nat. Commun.

During the revised process, we carried out time-dependent DFT (TD-DFT) calculations to investigate the in-depth mechanism and proved the universality of this MIEE strategy by several other macrocycles with different luminophores, therefore comprehensively improving the quality of our manuscript (Refer to "Reply to Reviewer #1"). For future work, we are going to explore further applications based on

MIEE.

Reply to Reviewer #3

Comments: Organic luminescent materials have drawn much attention in the past decades. Some effective methods have been built to enhance the emission efficiency, for example aggregation induced emission and crystallization induced emission. In this manuscript, authors developed a novel strategy, namely macrocyclization-induced emission enhancement (MIEE), for the improvement of luminophore's solid-state emission. A benzothiadiazole-based macrocycle (BT-LC) was found to show much more intense fluorescence in the solid state ($\Phi_{\text{PL}} = 99\%$) and exhibit better device performance in the application of OLEDs, in comparison with the corresponding monomer. Moreover, BT-LC showed strong emission in both dilute solution and aggregated state. MIEE is a new and effective approach to improve the luminophore's emission and would be helpful to developing organic luminescent materials. Hence, I believe that this work is highly suitable for publication after considering the following minor issues.

Reply: We greatly appreciate the reviewer's very positive comments. Herewith, we addressed the referee's comments as follows:

1. In assessment of solid-state photophysical properties, please add fluorescence lifetime of the monomers and the macrocycles reported.

Reply: According to the Reviewer's advice, the fluorescence lifetime of the monomer and the macrocycle has been added. The fluorescence lifetimes of BT-M in the solid state were measured as 8.45 ns (Figure R36), whereas BT-LC showed relatively longer lifetimes of 11.25 ns (Figure R37). Please see Page 5 in revised manuscript and Figure S36 and S37 in Supplementary Information.

The following discussion was added to the revised manuscript:

“The time-resolved emission decay properties of BT-M and BT-LC in the solid state were also studied. The fluorescence lifetimes of BT-M were measured as 8.45 ns, whereas BT-LC showed relatively longer lifetimes of 11.25 ns (Figure S36, S37).”

Figure R36 (Figure S35) PL decay spectra of BT-M in solid state.

Figure R37 (Figure S36) PL decay spectra of BT-LC in solid state.

2. For the synthesis, are there any other cyclic oligomers such as tetramer as the by-product?

Reply: There are no any other cyclic oligomers. The following discussion was added to the revised manuscript in Page 3:

“No other cyclic oligomers such as tetramer and pentamer were observed.”

3. In page 6, “BT-BC” should be BT-LC.

Reply: This was done.

4. Please define the abbreviated terms when they appear for the first time. “Tol” or “TOL”? Please be consistent.

Reply: The abbreviation was defined as “TOL”. The “Tol” was replaced by “TOL”.

References

1. Wang, Y., Xu, K., Li, B., Cui, L., Li, J., Jia, X., Zhao, H., Fang, J., Li, C., Efficient Separation of cis- and trans-1,2-Dichloroethene Isomers by Adaptive Biphen[3]arene Crystals. *Angew. Chem. Int. Ed.* **58**, 10281-10284 (2019).
2. Xu, K., Zhang, Z., Yu, C., Wang, B., Dong, M., Zeng, X., Gou, R., Cui, L., Li, C. J. A Modular Synthetic Strategy for Functional Macrocycles. *Angew. Chem. Int. Ed.* **59**, 7214-7218 (2020).
3. (a) Perdew, J. P., Burke, K., Ernzerhof, M. Generalized gradient approximation made simple. *Phys. Rev. Lett.* **77**, 3865-3868 (1996). (b) Perdew, J. P., Burke, K., Ernzerhof, M. Errata: Generalized gradient approximation made simple. *Phys. Rev. Lett.* **78**, 1396-1396 (1997). (c) Adamo, C., Barone, V. Toward reliable density functional methods without adjustable parameters: The PBE0 model. *J. Chem. Phys.* **110**, 6158-6169 (1999).
4. Frisch, M. J. *et al.* Gaussian 16 revision A.03 (Gaussian Inc., 2019).
5. Dreuw, A., Head-Gordon, M. Single-Reference ab Initio Methods for the Calculation of Excited States of Large Molecules. *Chem. Rev.* **105**, 4009-4037 (2005).
6. Matsika, S. Electronic Structure Methods for the Description of Nonadiabatic Effects and Conical Intersections. *Chem. Rev.* **121**, 9407-9449 (2021).
7. Koga, N., Morokuma, K. Determination of the lowest energy point on the crossing seam between two potential surfaces using the energy gradient. *Chem. Phys. Lett.* **119**, 371-374 (1985).
8. (a) Zhao, H., Bian, W., Liu, K. A theoretical study of the reaction of O(³P) with isobutene. *J. Phys. Chem. A* **110**, 7858-7866 (2006). (b) Zhao, S., Wu, W., Zhao, H., Wang, H., Yang, C., Liu, K., Su, H. Adiabatic and nonadiabatic reaction pathways of the O(³P) with propyne. *J. Phys. Chem. A* **113**, 23-34 (2009). (c) Liu, K., Li, Y., Su, J., Wang, B. The reliability of DFT methods to predict electronic structures and minimum energy crossing point for [Fe^{IV}O](OH)₂ models: A comparison study with MCQDPT method. *J. Comput. Chem.* **35**, 703-710 (2014).

- (d) Li, H., Li, D., Zeng, X., Liu, K., Beckers, H., Schaefer, H. F. III., Esselman, B. J., McMahon, R. J. Toward Understanding the Decomposition of Carbonyl Diazide $(\text{N}_3)_2\text{C}=\text{O}$ and Formation of Diazirone $\text{cyc}l\text{-N}_2\text{CO}$: Experiment and Computations *J. Phys. Chem. A* **119**, 8903-8911 (2015). (e) Wu, Z., Feng, R., Li, H., Xu, J., Deng, G., Abe, M., Begue, D., Liu, K., Zeng, X Fast Heavy-Atom Tunneling in Trifluoroacetyl Nitrene. *Angew. Chem. Int. Ed.* **56**, 15672-15676 (2017).
9. CYLview20; Legault, C. Y., Université de Sherbrooke, 2020 (<http://www.cylview.org>)
 10. Peng, Z., Zhang, K., Huang, Z., Wang, Z., Duttwyler, S., Wang, Y., Lu, P., Emissions from a triphenylamine–benzothiadiazole–monocarbaborane triad and its applications as a fluorescent chemosensor and a white OLED component. *J Mater Chem C*, **7**, 2430-2435 (2019).
 11. Sun, X., Xu, X., Qiu, W., Yu, G., Zhang, H., Gao, X., Chen, S., Song, Y., Liu, Y., A non-planar pentaphenylbenzene functionalized benzo[2,1,3]thiadiazole derivative as a novel red molecular emitter for non-doped organic light-emitting diodes. *J Mater Chem*, **18**, 2709 (2008).
 12. Zhao, Z., Deng, C., Chen, S., Lam, JWY., Qin, W., Lu, P., Wang, Z., Kwok, HS., Ma, Y., Qiu, H., Tang, BZ., Full emission color tuning in luminogens constructed from tetraphenylethene, benzo-2,1,3-thiadiazole and thiophene building blocks. *Chem Commun*, **47**, 8847-8849 (2011).
 13. Lee, WWH., Zhao, Z., Cai, Y., Xu, Z., Yu, Y., Xiong, Y., Kwok, RTK., Chen, Y., Leung, NLC., Ma, D., Lam, JWY., Qin, A., Tang, BZ., Facile access to deep red/near-infrared emissive AIEgens for efficient non-doped OLEDs. *Chem Sci*, **9**, 6118-6125 (2018).
 14. Li, Y., Wang, W., Zhuang, Z., Wang, Z., Lin, G., Shen, P., Chen, S., Zhao, Z., Tang, BZ., Efficient red AIEgens based on tetraphenylethene: synthesis, structure, photoluminescence and electroluminescence. *J Mater Chem C*, **6**, 5900-5907 (2018).
 15. Thangthong, A., Prachumrak, N., Sudyoasuk, T., Namuangruk, S., Keawin, T.,

- Jungsuttiwong, S., Kungwan, N., Promarak, V., Multi-triphenylamine–functionalized dithienylbenzothiadiazoles as hole-transporting non-doped red emitters for efficient simple solution processed pure red organic light-emitting diodes. *Org Electron*, **21**, 117-125 (2015).
16. Angioni, E., Chapran, M., Ivaniuk, K., Kostiv, N., Cherpak, V., Stakhira, P., Lazauskas, A., Tamulevius, S., Volyniuk, D., Findlay, N.J., Tuttle, T., Grazulevicius, J.V., Skabara, P.J., A single emitting layer white OLED based on exciplex interface emission. *J Mater Chem C*, **4**, 3851-3856 (2016).
17. Pathak, A., Justin Thomas, K.R., Singh, M., Jou, J.H., Fine-Tuning of Photophysical and Electroluminescence Properties of Benzothiadiazole-Based Emitters by Methyl Substitution. *J Org Chem*, **82**, 11512-11523 (2017).
18. Guo, J., Li, X.L., Nie, H., Luo, W., Gan, S., Hu, S., Hu, R., Qin, A., Zhao, Z., Su, S.J., Tang, B.Z., Achieving High-Performance Nondoped OLEDs with Extremely Small Efficiency Roll-Off by Combining Aggregation-Induced Emission and Thermally Activated Delayed Fluorescence *Adv Funct Mater*, **27**, 1606458 (2017).

REVIEWER COMMENTS

Reviewer #1 (Remarks to the Author):

Although the authors showed considerable efforts to address the reviewers' comments in their revised submission, this reviewer remains hesitating to recommend for publication in Nat. Comm., due to the following.

1. The main concept of "macrocyclization-induced emission enhancement (MIEE)" is still not fully validated. For comparison, the well known aggregation- or vibration-induced emissions focus on the fluorescence changes of same molecules under different states. This manuscript, however, is comparing fluorescence between different ones, namely macrocycles and their synthetic precursors (fractional parts). Therefore, although I consent with the experimental and computational results, the authors' comparisons seem somewhat unfair. What I am also wondering is the role of methylene linkers in the macrocycles which may cause certain molecular strains and "restrict intramolecular motions" as the authors mentioned. What if all or part of the methylenes are replaced by more flexible and extended linkers?

2. The OLED performance is not quite exciting. When talking about any types of well developed devices, readers of high-profile journals would expect to see something significant.

Reviewer #3 (Remarks to the Author):

Since reviewers' comments have been well addressed, the revised manuscript is recommended for publication.

Reply to Reviewer #1

Comments: Although the authors showed considerable efforts to address the reviewers' comments in their revised submission, this reviewer remains hesitating to recommend for publication in Nat. Comm., due to the following.

Reply: Thanks for the reviewer's comments. Herewith, we addressed the reviewer's comments as follows:

1. The main concept of "macrocyclization-induced emission enhancement (MIEE)" is still not fully validated. For comparison, the well known aggregation- or vibration-induced emissions focus on the fluorescence changes of same molecules under different states. This manuscript, however, is comparing fluorescence between different ones, namely macrocycles and their synthetic precursors (fractional parts). Therefore, although I consent with the experimental and computational results, the authors' comparisons seem somewhat unfair. What I am also wondering is the role of methylene linkers in the macrocycles which may cause certain molecular strains and "restrict intramolecular motions" as the authors mentioned. What if all or part of the methylenes are replaced by more flexible and extended linkers?

Reply: Thanks a lot for consenting with our experimental and computational results. This reviewer's previous great comments and advices, especially for the scope and mechanism of our strategy, help a lot for improving the quality of our manuscript. As a well-known strategy, aggregation-induced emission (AIE) provides an efficient approach to enhance emission and has been applied in many fields. Tetraphenyl ethylene (TPE) molecules are non-emissive in solution and highly emissive in aggregate (Figure R1). The essence is the restriction of intramolecular rotations of luminophore by aggregating. In most cases, the emission performance is compared between free single luminogen with aggregates which is composed of lots of luminogens. As shown in Figure R1, our MIEE strategy is also to restrict intramolecular motion of luminophore. The difference is utilizing the skeleton of macrocycle but not aggregating. The linker of methylene played the role of locking luminophores and would not disturb the emission of luminophores due to its non-conjugated character. Moreover, the computational results gave a reasonable

mechanism. With the double-bonding tendency, the non-radiative relaxation process requires that the torsion angle between benzothiadiazole and benzene ring can be twisted to near 20 degrees at $MECP_{S1/S0}$. That is to say, benzothiadiazole and benzene ring tend to be in the same plane at $MECP_{S1/S0}$. Unlike BT-M, BT-LC has a rigid triangular geometry and the rotation of the corresponding torsion angle would be limited (Figure 3a and S48). Therefore, BT-LC can avoid the process of $MECP_{S1/S0}$ non-radiative relaxation and its fluorescence efficiency is enhanced.

Considering that AIE is based on the aggregation of luminogens and our strategy is due to the cyclization of a few of luminogens by methylenes, we think “macrocyclization-induced emission enhancement (MIEE)” is an appropriate term. The following description was added to the introduction part of the revised manuscript: *“Since the emission enhancement is due to the cyclization of a few of luminogens by methylenes, it is termed as macrocyclization-induced emission enhancement (MIEE) (Scheme S1).”* And the following figure was added to the supporting information (Scheme S1).

Figure R1 (Scheme S1). Illustration of aggregation-induced emission (AIE) and macrocyclization-induced emission enhancement (MIEE).

It is an excellent suggestion to replace all or part of the methylenes by more

flexible and extended linkers. However, such synthesis is extremely difficult and beyond our ability. Actually, macrocycles with functionalized skeletons are hard to synthesize, although functional substituents can be conveniently attached to the macrocycle periphery. Our recent work developed a general and modular method to customize functional macrocycles containing diverse skeletons that are connected by methylenes. This modular synthesis provides the convenient production of diverse fluorescence macrocycles, ensuring the availability and versatility of MIEE. However, we are very sorry that the methylenes cannot be replaced by more flexible and extended linkers in this system.

2. The OLED performance is not quite exciting. When talking about any types of well developed devices, readers of high-profile journals would expect to see something significant.

Reply: Certainly, compared with reported BT-based emitters, macrocycle BT-LC showed medium current efficiency, power efficiency and external quantum efficiency. Although the performance of the device is inferior to that of the current state-of-art, it is the first example of macrocyclic arene-based OLED. Macrocycles would be potentially applied in OLEDs considering the following two points: 1) our modular synthesis method could conveniently produce diverse fluorescence macrocycles; 2) MIEE is an efficient strategy to improve Φ_{PL} values of chromophores.” From this point, this work is believed to be merited in this high-profile journal.

Reply to Reviewer #3

Comments: Since reviewers' comments have been well addressed, the revised manuscript is recommended for publication.

Reply: Thanks for the reviewer's very positive comment.

REVIEWERS' COMMENTS

Reviewer #1 (Remarks to the Author):

After reading the authors' responses, my previous concerns remain unaddressed. Therefore, I cannot recommend the current manuscript for publication in Nature Communications.

Again, high standards should apply in order to establish a new concept or strategy with generality, like the "macrocyclization-induced emission enhancement (MIEE)". It is unfair to conclude "MIEE" just by comparing the emission intensities of DIFFERENT compounds. Although the experimental observations were not questioned here, the word "macrocyclization" includes much broader scope than the limited examples in this manuscript (such as the "macrocyclized" compounds with longer and more flexible linkers). If we follow the authors' logic of "MIEE", then there are numerous examples in the literature showing emission changes after certain reactions. Can anyone claim analogous terminology like "methylation-induced emission enhancement" or "deprotection-induced emission enhancement"?

I also insist that the current OLED performances seem low for Nature Communications. First-time report but with inferior results should be more suitable for a specialized journal.

In addition, how did the authors define "the current state-of-art" as EQEmax of 8.47% (Line 166 in the manuscript)? After checking the corresponding reference in the SI, I cannot find this number.

Reviewer #1 (Remarks to the Author):

In this manuscript, Li et al reported a benzothiadiazole-derived macrocycle. The macrocycle was prepared by well-developed chemistry through condensation of electron-rich arenes with paraformaldehyde. The authors also claimed a concept of macrocyclization-induced emission enhancement (MIEE) because this newly synthesized macrocycle was found more fluorescent than its precursor. Overall, the manuscript reflects a reasonable amount of solid lab work. However, considering the high standards of Nat. Comm., favorable recommendation cannot be made for this manuscript based on the following reasons.

1. The introduction of the concept “macrocyclization-induced emission enhancement (MIEE)” seems to be a major selling point of this work. However, the manuscript is not well-organized to explain the molecular design, scope and limitation, and mechanism of this concept.

What are the considerations of molecular designs to trigger the so-called MIEE? How many different fluorophores can this concept be applicable to? And photophysically why?

2. It is not rare for an organic compound to be strongly fluorescent both in solution and in solid state. What would be really interesting is the special photophysical mechanism behind the observations. Unfortunately, this manuscript did not provide in-depth mechanistic investigation.

It seems confusing that the authors characterize the MIEE of the macrocycle using the analogous experiments of AIE. Such experiments were employed to trigger the formation of aggregates, but not macrocyclizations.

3. The authors' efforts to construct OLED devices based on the macrocycle should be applauded. But how does the devices' performance compared to the current state-of-art?

Reviewer #2 (Remarks to the Author):

In this paper, the authors described the synthesis of a benzothiadiazole-based macrocycle (BT-LC) and the macrocyclization-induced emission enhancement (MIEE) in the solid-state. Compared with the monomer 4,7-bis(2,4-dimethoxyphenyl)-2,1,3-benzothiadiazole (BT-M), the macrocycle BT-LC produces much more intense fluorescence in the solid state ($\Phi_{\text{PL}}=99\%$) and exhibits better device performance in the application of OLEDs. The MIEE can be ascribed to the restriction of intramolecular motion and the alleviation of the concentration

quenching by the macrocyclic topological structure. Although the increased emission of the macrocycle in solid state or its AIE property is very interesting, the phenomenon is general. Additionally, the performance of the device based on the macrocycle is not pleasantly surprised. Thus, I'm not sure that the paper is enough novelty to publish in Nat. Commun. Apparently, one stunning application for the macrocycle with strong emission will be powerful to improve the quality of the manuscript.

Reviewer #3 (Remarks to the Author):

Organic luminescent materials have drawn much attention in the past decades. Some effective methods have been built to enhance the emission efficiency, for example aggregation induced emission and crystallization induced emission. In this manuscript, authors developed a novel strategy, namely macrocyclization-induced emission enhancement (MIEE), for the improvement of luminophore's solid-state emission. A benzothiadiazole-based macrocycle (BT-LC) was found to show much more intense fluorescence in the solid state ($\Phi_{\text{PL}} = 99\%$) and exhibit better device performance in the application of OLEDs, in comparison with the corresponding monomer. Moreover, BT-LC showed strong emission in both dilute solution and aggregated state. MIEE is a new and effective approach to improve the luminophore's emission and would be helpful to developing organic luminescent materials. Hence, I believe that this work is highly suitable for publication after considering the following minor issues.

1. In assessment of solid-state photophysical properties, please add fluorescence lifetime of the monomers and the macrocycles reported.
2. For the synthesis, are there any other cyclic oligomers such as tetramer as the by-product?
3. In page 6, "BT-BC" should be BT-LC.
4. Please define the abbreviated terms when they appear for the first time. "Tol" or "TOL"? Please be consistent.

Corrections and changes made in response to the referees' comments

Manuscript submitted to *Nature Communications*

Title: Synthesis and Macrocyclization-Induced Emission Enhancement of Benzothiadiazole-based Macrocycle

Manuscript ID: NCOMMS-21-36571C by Chunju, Li et. al

We sincerely thank the reviewers for their comments about our submitted article. Corrections and necessary changes have been made according to the reviewers' comments, and are explained as follows:

(Referees' comments: in black; Corrections made by the authors in response to the comments: in blue)

To Referee 1:

We greatly appreciate the reviewer's comments and advices. Herewith, we addressed the referee's comments as follows:

Question 1

The introduction of the concept "macrocyclization-induced emission enhancement (MIEE)" seems to be a major selling point of this work. However, the manuscript is not well-organized to explain the molecular design, scope and limitation, and mechanism of this concept.

What are the considerations of molecular designs to trigger the so-called MIEE? How many different fluorophores can this concept be applicable to? And photophysically why?

Response to the question 1 of referee 1

1.1 Molecular design

Such design of BT-LC is mainly considering the following two features: 1) macrocyclization can restrict intramolecular motion by locking chromophores into the skeleton of macrocycle to suppress non-radiation relaxation; 2) it spatially separates chromophores in a single macrocycle to eliminate the concentration quenching to a

certain degree. Please see the discussion in Paragraph 2 Page 2:

“Such macrocyclization-induced emission enhancement (MIEE) is theoretically feasible considering the following two features: on one hand, it spatially separates chromophores in a single macrocycle to eliminate the concentration quenching to a certain degree,³⁵⁻³⁸ on the other hand, it restricts intramolecular motion by locking its chromophores into the skeleton of macrocycle to suppress non-radiation relaxation.^{39,40} Herein, we report the synthesis of a benzothiadiazole-based macrocycle (BT-LC) with three methylene bridges, which exhibits high fluorescence quantum yield in the solid state, up to 99%, much higher than that of BT-M. Experiments and theoretical calculations demonstrated that the enhanced emission can be ascribed to the efficient suppression of non-radiative relaxation process.”

1.2 Scope and limitation

To expand the application scope of MIEE, two kinds of macrocycles were designed and synthesized: one is cyclic trimer with triangle shape (C1-C4), and the other is cyclic dimer with tetragonum shape (C5-C7). They were prepared according to our recently reported method, where linear monomer produces trimers, while V-shaped one tends to form dimer (Figures R1).^{1,2} Photophysical properties of these macrocycles and monomers were shown in Figure R2-8 and summarized in Table R1. Similar to BT-LC, the other triangular macrocycles (C1-C4) also showed macrocyclization-induced emission enhancement, especially for octafluorobenzene-based C4, whose Φ_{PL} value greatly increased from 12.7% to 65.2%. For tetragonal macrocycles C5-C7, their quantum yields were also improved compared with corresponding monomers. In particular, the Φ_{PL} value of C5 is 9.8 times than that of M5. The results showed that MIEE is an effective and general strategy to enhance solid-state emitters, although for macrocycles C3 and C7, emission improvement is not significant. Figure R1 and Table R1 were added in the revised manuscript as Figure 6 and Table 2, respectively; Figure R2-8 were added in the revised Supplementary Information as Supplementary Figure S38-44; Figure R9 was added in the revised Supplementary Information as Supplementary Figure S1.

Figure R10-33 were added in the revised Supplementary Information as Supplementary Figure 8-31.

The following discussion was added to the revised manuscript in Page 9:

“Application scope of MIEE. To expand the application scope of MIEE, two kinds of macrocycles were designed and synthesized: one is cyclic trimer with triangle shape (C1-C4), and the other is cyclic dimer with tetragonum shape (C5-C7). They were prepared according to our recently reported method, where linear monomer produces trimers, while V-shaped one tends to form dimer (Figures 6 and Supplementary Figure 1).³⁴ Photophysical properties of these macrocycles and monomers were shown in Supplementary Figure 38-44 and summarized in Table 2. Similar to BT-LC, the other triangular macrocycles (C1-C4) also showed macrocyclization-induced emission enhancement, especially for octafluorobenzene-based C4, whose Φ_{PL} value greatly increased from 12.7% to 65.2%. For tetragonal macrocycles C5-C7, their quantum yields were also improved compared with corresponding monomers. In particular, the Φ_{PL} value of C5 is 9.8 times than that of M5. The results showed that MIEE is an effective and general strategy to enhance solid-state emitters, although for macrocycles C3 and C7, emission improvement is not significant.”

Figure R1 (Figure 6). Chemical structure of other selected MIEE macrocycles.

Table R1 (Table 2). Solid-state photophysical properties of monomers and

corresponding macrocycles

	M1	C1	M2	C2	M3	C3	M4	C4	M5	C5	M6	C6	M7	C7
$\lambda_{\text{ex}}(\text{nm})$	300	308	300	332	325	332	310	320	350	405	340	300	300	310
$\lambda_{\text{em}}(\text{nm})$	355	384	384	384	403	408	366	392	468	450	361	373	390	449
$\Phi_{\text{PL}}(\%)$	5.51	19.2	15.3	53.0	34.4	37.5	12.7	65.2	0.6	5.9	12.1	17.5	21.9	28.2

λ_{ex} (nm): excitation maximum. λ_{em} (nm): fluorescence maximum. Φ_{PL} : absolute PL quantum yield.

Figure R2 (Supplementary Figure 38) PL spectra of C1 and M1 (Insets: photographs in solid state under 365 nm UV illuminations).

Figure R3 (Supplementary Figure 39) PL spectra of C2 and M2 (Insets: photographs in solid state under 365 nm UV illuminations).

Figure R4 (Supplementary Figure 40) PL spectra of C3 and M3 (Insets: photographs in solid state under 365 nm UV illuminations).

Figure R5 (Supplementary Figure 41) PL spectra of C4 and M4 (Insets: photographs in solid state under 365 nm UV illuminations).

Figure R6 (Supplementary Figure 42) PL spectra of C5 and M5 (Insets: photographs in solid state under 365 nm UV illuminations).

Figure R7 (Supplementary Figure 43) PL spectra of C6 and M6 (Insets: photographs in solid state under 365 nm UV illuminations).

Figure R8 (Supplementary Figure 44) PL spectra of C7 and M7 (Insets: photographs in solid state under 365 nm UV illuminations).

Synthesis characterization of new compounds

Figure R9 (Supplementary Figure 1) Chemical structure of other monomers and corresponding macrocycles.

M4. Under the protection of N₂ atmosphere, 4,4'-dibromooctafluorobiphenyl (4.56 g, 10.0 mmol), 2,4-dimethoxyphenylboronic acid (4.55 g, 25.0 mmol), and tetrakis(triphenylphosphine)palladium(0) (1.16 g, 1.00 mmol) were dissolved in tetrahydrofuran (160 mL). The sodium carbonate (4.24 g, 40.0 mmol) in water (25 mL) was added into the solution and stirred for 24 h at 85 °C. Upon cooling to room temperature, water (120 mL), dichloromethane (120 mL) was added and stirred. After filtration of the solution, the solution was partitioned between dichloromethane and water. The product was extracted from the organic layer. The aqueous layer was further extracted twice with dichloromethane (120 mL). The combined organic layer was dried over anhydrous Na₂SO₄ and evaporated under reduced pressure. The product was purified by column chromatography on silica gel (eluent: 4/1, v/v, dichloromethane : petroleum ether) to give product M4 (4.33g, 76%) as a white solid. m.p. 198-199 °C; ¹H NMR (400 MHz, CDCl₃, 298 K) δ 7.25 (d, *J* = 8.0 Hz, 2H), 6.66-6.63 (m, 4H), 3.89 (s, 6H), 3.85 (s, 6H). ¹³C NMR (100 MHz, CDCl₃, 298 K) δ 162.5, 158.5, 145.5, 143.5, 132.4, 119.7, 108.5, 106.1, 105.1, 99.1, 55.9, 55.6. HRMS (ESI) *m/z*: [M+H]⁺ calcd for [C₂₈H₁₉F₈O₄]⁺, 571.1150; found, 571.1156.

C4. To the solution of M4 (2.85g, 5.00 mmol) in dichloromethane (200 mL) was added paraformaldehyde (0.450 g, 15.0 mmol). Boron trifluoride diethyl etherate (0.650 ml, 5.00 mmol) was then added to the reaction mixture. The mixture was stirred at 25 °C for 30 minutes. Then the reaction was quenched by addition of 200 mL saturated aqueous NaHCO₃. The solution was partitioned between dichloromethane and saturated aqueous NaHCO₃. The product was extracted from the

organic layer. The aqueous layer was further extracted twice with dichloromethane (120 mL). The combined organic layer was dried over anhydrous Na_2SO_4 and concentrated. The product was purified by column chromatography on silica gel (eluent : dichloromethane: ethyl acetate = 3:1) to obtain product C4 (1.05 g, 36%) as a white solid. m.p. $>320^\circ\text{C}$; ^1H NMR (400 MHz, DMSO, 353 K) δ 8.39-6.68 (m, 12H), 3.92 (s, 18H), 3.86 (s, 18H), 3.81 (s, 6H). ^{13}C NMR (100 MHz, CDCl_3 , 298 K) δ 159.8, 156.9, 145.6, 143.1, 132.5, 131.0, 128.9, 121.4, 119.7, 107.4, 105.9, 95.7, 56.0, 55.8, 28.1. HRMS (ESI) m/z: $[\text{M}+\text{H}]^+$ calcd for $[\text{C}_{87}\text{H}_{55}\text{F}_{24}\text{O}_{12}]^+$, 1747.3305; found, 1747.3251.

M5. Under the protection of N_2 atmosphere, 4,4'-dibromobenzophenone (6.80 g, 20.0 mmol), 2,4-dimethoxybenzeneboronic acid (9.10 g, 50.0 mmol), and tetrakis(triphenylphosphine)palladium(0) (2.32 g, 2.00 mmol) were dissolved in tetrahydrofuran (350 mL). The sodium carbonate (8.48 g, 80.0 mmol) in water (50 mL) was added into the solution and stirred for 24 h at 85°C . Upon cooling to room temperature, water (200 mL), dichloromethane (200 mL) was added and stirred. After filtration of the solution, the solution was partitioned between dichloromethane and water. The product was extracted from the organic layer. The aqueous layer was further extracted twice with dichloromethane (200 mL). The combined organic layer was dried over anhydrous Na_2SO_4 and evaporated under reduced pressure. The product was purified by column chromatography on silica gel (eluent: 4/1, v/v, dichloromethane : petroleum ether) to give product M5 (8.54g, 94%) as a white solid. m.p. $175\text{-}176^\circ\text{C}$; ^1H NMR (600 MHz, CDCl_3 , 298 K) δ 7.90 (d, $J = 6.00$ Hz, 4H),

7.65 (d, $J = 6.00$ Hz, 4H), 7.32 (d, $J = 6.00$ Hz, 2H), 6.62-6.59 (m, 4H), 3.87 (s, 6H), 3.84 (s, 6H). ^{13}C NMR (150 MHz, CDCl_3 , 298 K) δ 196.3, 161.0, 157.7, 142.8, 135.8, 131.5, 130.0, 129.3, 122.5, 105.0, 99.2, 55.7, 55.6. HRMS (ESI) m/z : $[\text{M}+\text{H}]^+$ calcd for $[\text{C}_{29}\text{H}_{27}\text{O}_5]^+$, 455.1853; found, 455.1860.

C5. To the solution of M5 (0.910 g, 2.00 mmol) in chloroform (100 mL) was added paraformaldehyde (0.180 g, 6.00 mmol). Boron trifluoride diethyl etherate (0.260 mL, 2.00 mmol) was then added to the reaction mixture. The mixture was stirred at 25 °C for 3 hours. Then the reaction was quenched by addition of 100 mL saturated aqueous NaHCO_3 . The solution was partitioned between dichloromethane and saturated aqueous NaHCO_3 . The product was extracted from the organic layer. The aqueous layer was further extracted twice with dichloromethane (100 mL). The combined organic layer was dried over anhydrous Na_2SO_4 and concentrated. The product was purified by column chromatography on silica gel (eluent : dichloromethane: ethyl acetate = 8:1) to obtain product C5 (0.056 g, 6%) as a white solid. m.p. $>320^\circ\text{C}$; ^1H NMR (600 MHz, CDCl_3 , 298 K) δ 7.81 (d, $J = 12.0$ Hz, 8H), 7.57 (d, $J = 6.00$ Hz, 8H), 7.03 (s, 4H), 6.57 (s, 4H), 3.94 (s, 4H), 3.90 (s, 12H), 3.84 (s, 12H). ^{13}C NMR (150 MHz, CDCl_3 , 298 K) δ 196.5, 158.4, 156.1, 142.9, 135.6, 131.9, 130.1, 129.3, 121.7, 121.3, 96.0, 56.1, 55.9, 27.3. HRMS (ESI) m/z : $[\text{M}+\text{H}]^+$ calcd for $[\text{C}_{60}\text{H}_{53}\text{O}_{10}]^+$, 933.3633; found, 933.3648.

M6. Under the protection of N_2 atmosphere, 3,5-dibromobenzonitrile (5.22 g, 20.0 mmol), 2,4-dimethoxybenzeneboronic acid (9.10 g, 50.0 mmol), and tetrakis(triphenylphosphine)palladium(0) (2.32 g, 2.00 mmol) were dissolved in

tetrahydrofuran (300 mL). The sodium carbonate (6.36 g, 60.0 mmol) in water (40 mL) was added into the solution and stirred for 24 h at 85 °C. Upon cooling to room temperature, water (150 mL), dichloromethane (150 mL) was added and stirred. After filtration of the solution, the solution was partitioned between dichloromethane and water. The product was extracted from the organic layer. The aqueous layer was further extracted twice with dichloromethane (150 mL). The combined organic layer was dried over anhydrous Na₂SO₄ and evaporated under reduced pressure. The product was purified by column chromatography on silica gel (eluent: 4/1, v/v, dichloromethane : petroleum ether) to give product M5 (6.83 g, 91%) as a white solid. m.p. 185-186 °C; ¹H NMR (400 MHz, CDCl₃, 298 K) δ 7.82 (s, 1H), 7.74 (s, 2H), 7.26 (d, *J* = 8.0 Hz, 2H), 6.60-6.58 (m, 4H), 3.87 (s, 6H), 3.83 (s, 6H). ¹³C NMR (100 MHz, CDCl₃, 298 K) δ 161.1, 157.5, 139.3, 134.9, 131.3, 131.1, 121.4, 119.7, 111.7, 105.0, 99.0, 55.7, 55.6. HRMS (ESI) *m/z*: [M+H]⁺ calcd for [C₂₃H₂₂NO₄]⁺, 376.1543; found, 376.1540.

C6. To the solution of M6 (1.86 g, 5.0 mmol) in 1,2-dichloroethane (150 mL) was added paraformaldehyde (0.450 g, 15.00 mmol). Boron trifluoride diethyl etherate (0.650 ml, 5.00 mmol) was then added to the reaction mixture. The mixture was stirred at 25 °C for 45 minutes. Then the reaction was quenched by addition of 150 mL saturated aqueous NaHCO₃. The solution was partitioned between dichloromethane and saturated aqueous NaHCO₃. The product was extracted from the organic layer. The aqueous layer was further extracted twice with dichloromethane (150 mL). The combined organic layer was dried over anhydrous Na₂SO₄ and concentrated. The product was purified by column chromatography on silica gel (eluent : dichloromethane: ethyl acetate = 20:1) to obtain product C6 (0.213 g, 11%) as a white solid. m.p. >320°C; ¹H NMR (400 MHz, CDCl₃, 298 K) δ 7.83 (s, 4H), 7.26 (s, 2H), 6.77 (s, 4H), 6.56 (s, 4H), 3.88 (s, 12H), 3.86 (s, 12H), 3.82 (s, 4H). ¹³C NMR (100 MHz, CDCl₃, 298 K) δ 158.6, 155.7, 139.9, 134.0, 131.5, 131.4, 121.0, 119.9, 110.9, 95.3, 55.8, 55.7, 28.5. HRMS (ESI) *m/z*: [M+Na]⁺ calcd for [C₄₈H₄₂N₂O₈]⁺, 797.2833; found, 797.2839.

M7. Under the protection of N_2 atmosphere, 1,3-bis(5-bromopyridin-2-yl)benzene (3.88 g, 10.0 mmol), 2,4-dimethoxybenzeneboronic acid (5.46 g, 30.0 mmol), and tetrakis(triphenylphosphine)palladium(0) (1.16 g, 1.00 mmol) were dissolved in tetrahydrofuran (200 mL). The sodium carbonate (3.18 g, 30.0 mmol) in water (30 mL) was added into the solution and stirred for 24 h at 85 °C. Upon cooling to room temperature, water (120 mL), dichloromethane (120 mL) was added and stirred. After filtration of the solution, the solution was partitioned between dichloromethane and water. The product was extracted from the organic layer. The aqueous layer was further extracted twice with dichloromethane (120 mL). The combined organic layer was dried over anhydrous Na_2SO_4 and evaporated under reduced pressure. The product was purified by column chromatography on silica gel (eluent: 5/1, v/v, dichloromethane : ethyl acetate) to give product M5 (4.48 g, 89%) as a pale yellow solid. m.p. 213-214; 1H NMR (400 MHz, $CDCl_3$, 298 K) δ 8.87 (s, 2H), 8.70 (s, 1H), 8.11 (d, $J = 8.0$ Hz, 2H), 7.95-7.87 (m, 4H), 7.60 (s, 1H), 7.33 (d, $J = 12.0$ Hz, 2H), 6.64-6.60 (m, 4H), 3.87 (s, 6H), 3.84 (s, 6H). ^{13}C NMR (100 MHz, $CDCl_3$, 298 K) δ 161.4, 158.2, 155.3, 150.4, 140.2, 137.7, 133.0, 131.4, 129.6, 127.6, 125.7, 120.3, 120.1, 105.4, 99.5, 56.0, 55.9. HRMS (ESI) m/z: $[M+H]^+$ calcd for $[C_{32}H_{29}N_2O_4]^+$, 505.2122; found, 505.2130.

C7. To the solution of M7 (1.08 g, 2.0 mmol) in 1,2-dichloroethane (200 mL) was added paraformaldehyde (0.180 g, 6.00 mmol). Boron trifluoride diethyl etherate (0.260 ml, 2.00 mmol) was then added to the reaction mixture. The mixture was stirred at 25 °C for 5 hours. Then the reaction was quenched by addition of 150 mL

saturated aqueous NaHCO₃. The solution was partitioned between dichloromethane and saturated aqueous NaHCO₃. The product was extracted from the organic layer. The aqueous layer was further extracted twice with dichloromethane (150 mL). The combined organic layer was dried over anhydrous Na₂SO₄ and concentrated. The product was purified by column chromatography on silica gel (eluent : dichloromethane: ethyl acetate = 1:1) to obtain product C7 (0.248 g, 24%) as a white solid. m.p. >320°C; ¹H NMR (400 MHz, CDCl₃, 298 K) δ 8.81 (s, 4H), 8.66 (s, 2H), 8.07 (d, *J* = 8.0 Hz, 4H), 7.90-7.86 (m, 8H), 7.58 (s, 2H), 6.99 (s, 4H), 6.59 (s, 4H), 3.93 (s, 4H), 3.90 (s, 12H), 3.84 (s, 12H). ¹³C NMR (100 MHz, CDCl₃, 298 K) δ 158.6, 156.3, 154.9, 150.2, 140.0, 137.5, 133.0, 131.7, 129.2, 127.1, 125.7, 121.8, 120.2, 120.0, 119.1, 105.3, 99.3, 96.1, 56.0, 55.4, 27.6. HRMS (ESI) *m/z*: [M+H]⁺ calcd for [C₆₆H₅₇N₄O₈]⁺, 1033.4171; found, 1033.4171.

Figure R10 (Supplementary Figure 8) ¹H NMR spectrum (400 MHz, CDCl₃, 298 K) of M4.

Figure R11 (Supplementary Figure 9) ^{13}C NMR spectrum (100 MHz, CDCl_3 , 298 K) of M4.

Figure R12 (Supplementary Figure 10). HMRS spectrum of M4.

Figure R13 (Supplementary Figure 11). ¹H NMR spectrum (400 MHz, DMSO, 353K) of C4. Due to the isomerization, the protons of C4 is complicated even at high temperature (353K). However, the HRMS confirmed it in Supplementary Figure 15.

Figure R14 (Supplementary Figure 12) ¹³C NMR spectrum (100 MHz, CDCl₃, 298 K) of C4.

Figure R15 (Supplementary Figure 13) HMRS spectrum of C4.

Figure R16 (Supplementary Figure 14) ¹H NMR spectrum (600 MHz, CDCl₃, 298 K) of M5.

Figure R17 (Supplementary Figure 15) ¹³C NMR spectrum (150 MHz, CDCl₃, 298 K) of M5.

Figure R18 (Supplementary Figure 16) HRMS spectrum of M5.

Figure R19 (Supplementary Figure 17) ¹H NMR spectrum (600 MHz, CDCl₃, 298 K) of C5.

Figure R20 (Supplementary Figure 18) ¹³C NMR spectrum (150 MHz, CDCl₃, 298 K) of M5.

Figure R21 (Supplementary Figure 19) HRMS spectrum of C5.

Figure R22 (Supplementary Figure 20) ^1H NMR spectrum (400 MHz, CDCl_3 , 298 K) of M6.

Figure R23 (Supplementary Figure 21) ^{13}C NMR spectrum (100 MHz, CDCl_3 , 298 K) of M6.

Figure R24 (Supplementary Figure 22) HMRS spectrum of M6.

Figure R25 (Supplementary Figure 23) ¹H NMR spectrum (400 MHz, CDCl₃, 298 K) of C6 (* = petroleum ether peak signals).

Figure S26 (Supplementary Figure 24) ¹³C NMR spectrum (100 MHz, CDCl₃, 298 K) of C6 (* = petroleum ether peak signals).

Figure R27 (Supplementary Figure 25) HMRS spectrum of C6.

Figure R28 (Supplementary Figure 26) ¹H NMR spectrum (400 MHz, CDCl₃, 298 K) of M7 (* = ethyl acetate peak signals).

Figure R29 (Supplementary Figure 27) ¹³C NMR spectrum (100 MHz, CDCl₃, 298 K) of M7 (* = ethyl acetate peak signals).

Figure R30 (Supplementary Figure 28) HMRMS spectrum of M7.

Figure R31 (Supplementary Figure 29) ¹H NMR spectrum (400 MHz, CDCl₃, 298 K) of C7 (* = ethyl acetate peak signals).

Figure R32 (Figure S30) ¹³C NMR spectrum (100 MHz, CDCl₃, 298 K) of C7 (* = ethyl acetate peak signals).

Figure R33 (Supplementary Figure 31) HMRS spectrum of C7.

1.3 Mechanism of this concept

To further illustrate the principle of MIEE, we calculated the process of radiative relaxation and non-radiative relaxation. As a competition of luminescence process, nonradiative relaxation process is semi-quantitatively described using the TDDFT method. As shown in the Figure R34a, BT-M can return to the ground state through a MECP_{S_1/S_0} , resulting in the decrease of fluorescence efficiency. It is particularly noteworthy that the C-C bond between benzothiadiazole and adjacent phenyl rings is gradually shortened according to the order of $S_{0\text{min}}$, $S_{1\text{min}}$ and MECP_{S_1/S_0} . Among them, the C-C bond length in MECP_{S_1/S_0} is only 1.41 Å (Figure R34b), which is distinctly shorter than the C-C single bond. With the double-bonding tendency, the non-radiative relaxation process requires that the torsion angle between benzothiadiazole and benzene ring can be twisted to near 20 degrees at MECP_{S_1/S_0} . That is to say, benzothiadiazole and benzene ring tend to be in the same plane at MECP_{S_1/S_0} . Unlike BT-M, BT-LC has a rigid triangular geometry and the rotation of the corresponding torsion angle would be limited (Figure 3a and S48). Therefore, BT-LC can avoid the process of MECP_{S_1/S_0} non-radiative relaxation and therefore its fluorescence efficiency is enhanced. These calculation results are consistent with our

assumption at the beginning of the article. It should be pointed out that the calculation of BT-LC was not performed due to its too many atoms for this calculation method. Figure R34 was added in the revised manuscript as Figure 4.

The following discussion was added to the revised manuscript in Page 6:

“Mechanism study of MIEE. To further illustrate the principle of MIEE, we calculated the process of radiative relaxation and non-radiative relaxation. As a competition of luminescence process, non-radiative relaxation process is semi-quantitatively described using the TDDFT method. As shown in the Figure 4a, BT-M can return to the ground state through a $MECP_{S1/S0}$, resulting in the decrease of fluorescence efficiency. It is particularly noteworthy that the C-C bond between benzothiadiazole and adjacent phenyl rings is gradually shortened according to the order of $S0_{min}$, $S1_{min}$ and $MECP_{S1/S0}$. Among them, the C-C bond length in $MECP_{S1/S0}$ is only 1.41 Å (Figure 4b), which is distinctly shorter than the C-C single bond. With the double-bonding tendency, the non-radiative relaxation process requires that the torsion angle between benzothiadiazole and benzene ring can be twisted to near 20 degrees at $MECP_{S1/S0}$. That is to say, benzothiadiazole and benzene ring tend to be in the same plane at $MECP_{S1/S0}$. Unlike BT-M, BT-LC has a rigid triangular geometry and the rotation of the corresponding torsion angle would be limited (Figure 3a and Supplementary Figure 48). Therefore, BT-LC can avoid the process of $MECP_{S1/S0}$ non-radiative relaxation and its fluorescence efficiency is enhanced. These calculation results are consistent with our assumption at the beginning of the article. It should be pointed out that the calculation of BT-LC was not performed due to its too many atoms for this calculation method.”

Figure R34 (Figure 4) Radiative and non-radiative relaxation process of BT-M calculated at TDA-PBE0/PBE0/6-31G* level. **a** the non-radiative relaxation process of BT-M via minimum energy crossing point (MECP_{S₁/S₀); **b** the minimum energy structures of BT-M in ground state ($S_{0_{min}}$), singlet state ($S_{1_{min}}$), and minimum energy crossing point (MECP_{S₁/S₀). The selected bond lengths are in Å and the selected torsion angles are in degree.}}

Quantum chemical calculation methods

All the calculations of ground states were performed at the PBE0/6-31g* level³ using the Gaussian16 suite of programs.⁴ For excited state calculation, The Tamm-Dancoff approximation (TDA)⁵ is used for TDDFT because it is more stable near minimum energy crossing point (MECP).⁶ Harmonic vibration frequency calculations are used to confirm the stationary points. MECP_{S₁/S₀} is located at the TDA-PBE0/PBE0/6-31G* level using the Newton-Lagrange method, which was introduced by Koga and Morokuma.⁷ These calculations are treated using a homemade program LookForMECP (version 2.1). This program can be obtained from the authors upon request. The early version of this program has been used successfully to search the MECP.⁸ The 3D figures of molecular structure were prepared by CYLView.⁹

Coordinates (Å) and energies (Hartree)

S0_{min}

E = -1657.336726 hartree

C	0.62416600	-1.29102100	0.13084600
C	-0.71323000	-1.28214100	0.20256300
C	1.39082400	-0.06492000	0.36866700
C	-1.40919900	-0.00794700	0.21488500
N	-1.36483300	-2.59721700	0.10475600
N	1.22798300	-2.62410200	-0.04674200
C	0.67565300	1.16800000	0.62509200
C	2.79846900	-0.03149100	0.25108400
C	-0.66501200	1.19953200	0.51767500
C	-2.80061900	0.08126200	-0.04765500
S	-0.02073200	-3.31610100	0.80461600
H	1.21697800	2.07610500	0.83790400
C	3.44881900	-1.11635000	-0.43584200
C	3.67073400	1.06531900	0.66854100
H	-1.19538500	2.13039600	0.65709400
C	-3.61927300	-1.07960900	0.15523900
C	-3.49446800	1.29135300	-0.46523400
C	4.73678400	-1.03446200	-0.88981900
H	2.83104100	-2.01221500	-0.56714900
C	4.96344900	1.14495100	0.19862700

O	3.18121500	1.94252800	1.54982700
C	-4.98892000	-1.03095000	0.11140800
H	-3.05786900	-1.99865000	0.36724700
C	-4.87196200	1.33375500	-0.49557300
O	-2.74028400	2.32021700	-0.87209600
C	5.49402500	0.11940100	-0.60665000
H	5.16577200	-1.86579200	-1.43658400
H	5.62398600	1.96114500	0.46141400
C	4.01265100	2.98933300	2.01473200
C	-5.62573700	0.18547500	-0.19058200
H	-5.56150900	-1.92894400	0.31156100
H	-5.41660200	2.22126300	-0.79129300
C	-3.37640900	3.49641100	-1.33215400
O	6.74835300	0.31466000	-1.00323800
H	4.90246100	2.58835500	2.51331900
H	4.31419200	3.65234800	1.19534300
H	3.40941100	3.54473800	2.73286700
O	-6.94662600	0.35481700	-0.24799300
H	-4.00478800	3.28499500	-2.20520500
H	-3.98343500	3.95750500	-0.54399500
H	-2.57065600	4.17321500	-1.61725400
C	7.38520100	-0.67865700	-1.79692700
C	-7.79300100	-0.75846800	0.00129400
H	7.47975400	-1.61850000	-1.24405200
H	6.83547700	-0.84653100	-2.72829700
H	8.37558700	-0.28239900	-2.01982400
H	-7.62178300	-1.55315200	-0.73194000
H	-7.64153000	-1.14732700	1.01350200
H	-8.81037200	-0.38012600	-0.09791500

S1_{min}

E = -1657.235636 hartree

C	1.42952000	-0.60458200	0.23031300
C	0.69347800	-1.66353500	0.79348900
C	0.71473200	0.45474800	-0.41033000
C	-0.69331400	-1.66364200	0.79352600
H	1.22788500	-2.46228500	1.30045400
C	2.88463000	-0.66117100	0.23172800
C	-0.71492200	0.45464700	-0.41027800
N	1.28355800	1.44487000	-1.12789000
C	-1.42952400	-0.60477500	0.23041400
H	-1.22760000	-2.46248000	1.30048600
C	3.55286200	-1.85321500	-0.08548900
C	3.69649400	0.47944500	0.52599600
N	-1.28397100	1.44467700	-1.12777500
S	-0.00029400	2.32234000	-1.74064700
C	-2.88461200	-0.66143300	0.23187600
C	4.92961700	-1.92752000	-0.21119200
H	2.95281700	-2.73112500	-0.30627600
C	5.07615100	0.42854200	0.37256300
O	3.04811300	1.53798000	1.01199800
C	-3.55296100	-1.85343100	-0.08521600
C	-3.69638000	0.47933000	0.52604500

C	5.69477000	-0.76670500	0.00008100
H	5.39562500	-2.86283500	-0.49843200
H	5.70597300	1.28721100	0.57182100
C	3.70339500	2.78642800	1.03737000
C	-4.92971600	-1.92753800	-0.21099500
H	-2.95301600	-2.73144500	-0.30586400
C	-5.07602900	0.42863300	0.37244300
O	-3.04788300	1.53770100	1.01217800
O	7.03623100	-0.71415500	-0.11061500
H	4.51966700	2.79647200	1.77148100
H	4.09021900	3.04025800	0.04438400
H	2.94406100	3.51226500	1.32865300
C	-5.69475100	-0.76658100	0.00005200
H	-5.39584300	-2.86281600	-0.49816700
H	-5.70572100	1.28742700	0.57156700
C	-3.70277800	2.78634500	1.03727200
C	7.72968800	-1.89046900	-0.47065800
O	-7.03617600	-0.71391700	-0.11083100
H	-4.08935000	3.04017300	0.04418800
H	-4.51917000	2.79672300	1.77124900
H	-2.94327900	3.51200000	1.32857800
H	7.58291300	-2.68518200	0.27111900
H	7.41955800	-2.25105400	-1.45899300
H	8.78492200	-1.61603700	-0.50146000
C	-7.72977700	-1.89029400	-0.47040900
H	-7.41986000	-2.25117100	-1.45870400
H	-7.58290900	-2.68479800	0.27156800
H	-8.78499600	-1.61579300	-0.50108700

MECP_{S1/S0}

E = -1657.157554 hartree

C	0.62416600	-1.29102100	0.13084600
C	-0.71323000	-1.28214100	0.20256300
C	1.39082400	-0.06492000	0.36866700
C	-1.40919900	-0.00794700	0.21488500
N	-1.36483300	-2.59721700	0.10475600
N	1.22798300	-2.62410200	-0.04674200
C	0.67565300	1.16800000	0.62509200
C	2.79846900	-0.03149100	0.25108400
C	-0.66501200	1.19953200	0.51767500
C	-2.80061900	0.08126200	-0.04765500
S	-0.02073200	-3.31610100	0.80461600
H	1.21697800	2.07610500	0.83790400
C	3.44881900	-1.11635000	-0.43584200
C	3.67073400	1.06531900	0.66854100
H	-1.19538500	2.13039600	0.65709400
C	-3.61927300	-1.07960900	0.15523900
C	-3.49446800	1.29135300	-0.46523400
C	4.73678400	-1.03446200	-0.88981900
H	2.83104100	-2.01221500	-0.56714900
C	4.96344900	1.14495100	0.19862700
O	3.18121500	1.94252800	1.54982700
C	-4.98892000	-1.03095000	0.11140800

H	-3.05786900	-1.99865000	0.36724700
C	-4.87196200	1.33375500	-0.49557300
O	-2.74028400	2.32021700	-0.87209600
C	5.49402500	0.11940100	-0.60665000
H	5.16577200	-1.86579200	-1.43658400
H	5.62398600	1.96114500	0.46141400
C	4.01265100	2.98933300	2.01473200
C	-5.62573700	0.18547500	-0.19058200
H	-5.56150900	-1.92894400	0.31156100
H	-5.41660200	2.22126300	-0.79129300
C	-3.37640900	3.49641100	-1.33215400
O	6.74835300	0.31466000	-1.00323800
H	4.90246100	2.58835500	2.51331900
H	4.31419200	3.65234800	1.19534300
H	3.40941100	3.54473800	2.73286700
O	-6.94662600	0.35481700	-0.24799300
H	-4.00478800	3.28499500	-2.20520500
H	-3.98343500	3.95750500	-0.54399500
H	-2.57065600	4.17321500	-1.61725400
C	7.38520100	-0.67865700	-1.79692700
C	-7.79300100	-0.75846800	0.00129400
H	7.47975400	-1.61850000	-1.24405200
H	6.83547700	-0.84653100	-2.72829700
H	8.37558700	-0.28239900	-2.01982400
H	-7.62178300	-1.55315200	-0.73194000
H	-7.64153000	-1.14732700	1.01350200
H	-8.81037200	-0.38012600	-0.09791500

Question 2

It is not rare for an organic compound to be strongly fluorescent both in solution and in solid state. What would be really interesting is the special photophysical mechanism behind the observations. Unfortunately, this manuscript did not provide in-depth mechanistic investigation.

It seems confusing that the authors characterize the MIEE of the macrocycle using the analogous experiments of AIE. Such experiments were employed to trigger the formation of aggregates, but not macrocyclizations.

Response to the question 2 of referee 1

Thanks for reviewer. We agree the reviewer's opinion that characterizing the MIEE of the macrocycle by the analogous experiments of AIE is not suitable. Therefore, related discussion and characterization were moved to the Supplementary information. Please see Supplementary Figure 32, Supplementary Figure 35 and Supplementary Table 1 in the revised Supplementary Information.

To illustrate MIEE, we firstly performed the photoluminescence (PL) spectra. The enhanced fluorescence was observed in the PL spectra (Figure R35). The quantum yield (Φ_{PL}) for BT-LC (99%) is much higher than that for the monomer (65%). The results illustrate the effect of MIEE. Figure R35 was added in the revised manuscript as Figure 2. The following discussion was added to the revised manuscript in Page 4:

“As depicted in Figure 2, BT-LC exhibited a red-shifted emission ($\lambda_{em}=562$ nm) compared to BT-M ($\lambda_{em}=491$ nm). Also, enhanced fluorescence was observed in the photoluminescence (PL) spectra. The quantum yield (Φ_{PL}) for BT-LC (99%) is much higher than that for the monomer (65%).”

Figure R35 (Figure 2) PL spectra of BT-M and BT-LC in the solid state (Insets: photographs in solid state under 365 nm UV illuminations).

As mentioned above, we carried out time-dependent DFT (TD-DFT) calculations to investigate the in-depth mechanism. Please refer to “**1.3 mechanism of this concept**” (Page 25).

Question 3

The authors’ efforts to construct OLED devices based on the macrocycle should be applauded. But how does the devices’ performance compared to the current state-of-art?

Response to the question 3 of referee 1

Thanks for this great suggestion. We summarized the electroluminescent properties based on BT emitters in Table R2 and Figure R36. Among the BT-based OLEDs, the macrocycle (BT-LC) showed medium current efficiency (CE_{\max} , 9.93 cd A^{-1}), power efficiency (PE_{\max} , 8.25 lm W^{-1}) and external quantum efficiency (EQE_{\max} , 2.82%). To the best of our knowledge, this is the first report that macrocyclic arenes are used as emitters in OLEDs. The compound 20 exhibited the current state-of-art device performance and its CE_{\max} , PE_{\max} and EQE_{\max} reached 30.4 cdA^{-1} , 23.67 lmW^{-1} and 8.47%, respectively. (J. Mater. Chem. C, 2020, 8, 6851-6860). Figure R36 and Table R2 were added in the revised Supplementary Information as Supplementary Figure S56 and Table S4.

The following discussion was added to the revised manuscript in Page 8:

“Certainly, compared with reported BT-based emitters, macrocycle BT-LC showed moderate CE_{max} , PE_{max} and EQE_{max} (Supplementary Figure 56 and Supplementary Table 4). Although the performance of the device is inferior to that of the current state-of-art (EQE_{max} , 8.47%), it is the first example of macrocyclic arene-based OLED. Macrocycles would be potentially applied in OLEDs considering the following two points: 1) our modular synthesis method could conveniently produce diverse fluorescence macrocycles;³⁴ 2) MIEE is an efficient strategy to improve Φ_{PL} values of chromophores.”

Figure R36 (Supplementary Figure 56). Chemical structures of benzothiadiazole-based emitters.

Table R2 (Supplementary Table 4) Electroluminescence properties of benzothiadiazole-based emitters

	CE_{\max} (cd A^{-1})	PE_{\max} (lm W^{-1})	EQE_{\max} (%)	Ref.
BT-M	10.1	7.10	1.92	This work
BT-LC	9.93	8.25	2.82	This work
1	0.28	0.15	0.15	10
2	0.68	0.51	0.32	10
3	0.88	0.64	0.40	10
4	1.37	- ^a	1.00	11
5	0.4	0.5	1.00	12
6	5.2	3.0	1.50	12
7	6.4	2.9	3.10	12
8	- ^a	- ^a	1.43	13
9	- ^a	- ^a	1.73	13
10	1.31	1.59	2.17	14
11	1.41	1.70	2.03	14
12	2.19	1.61	2.09	14
13	6.25	5.17	- ^a	15
14	6.5	2.6	2.39	16
15	6.2	11.6	4.5	17
16	15.7	12.2	4.6	17
17	15.2	10.9	4.8	17
18	30.4	23.67	8.47	18

^a means that this value is not given in the original reference.

To Referee 2:

Question

In this paper, the authors described the synthesis of a benzothiadiazole-based macrocycle (BT-LC) and the macrocyclization-induced emission enhancement (MIEE) in the solid-state. Compared with the monomer 4,7-bis(2,4-dimethoxyphenyl)-2,1,3-benzothiadiazole (BT-M), the macrocycle BT-LC produces much more intense fluorescence in the solid state ($\Phi_{\text{PL}}=99\%$) and exhibits better device performance in the application of OLEDs. The MIEE can be ascribed to the restriction of intramolecular motion and the alleviation of the concentration quenching by the macrocyclic topological structure. Although the increased emission of the macrocycle in solid state or its AIE property is very interesting, but the phenomenon is general. Additionally, the performance of the device based on the macrocycle is not pleasantly surprised. Thus, I'm not sure that the paper is enough novelty to publish in *Nat. Commun.* Apparently, one stunning application for the macrocycle with strong emission will be powerful to improve the quality of the manuscript.

Response to question of referee 2

We greatly appreciate the reviewer's advices. Herewith, we addressed the referee's comments as follows:

Organic luminescent materials with high quantum efficiencies have attracted intensive attention. However, most of organic luminogens suffer from severe quenching effect in the aggregate state due to the formation of such detrimental aggregates as excimers and exciplexes, which greatly limits their applications in organic luminescent materials. We presented a novel strategy for the improvement of luminophore's solid-state emission, i.e., macrocyclization-induced emission enhancement (MIEE), by linking luminophores through $\text{C}(\text{sp}^3)$ bridges to give a macrocycle. We think this work has the novelty to be published in *Nat. Commun.*

During the revised process, we carried out time-dependent DFT (TD-DFT)

calculations to investigate the in-depth mechanism and proved universality of this MIEE strategy by seven other macrocycles with different luminophores, therefore comprehensively improving the quality of our manuscript (Refer to “Reply to Reviewer #1”). For future work, we are going to explore further applications based on MIEE.

To Referee 3:

We greatly appreciate the reviewer’s positive comments. Herewith, we addressed the referee’s comments as follows:

Question 1

In assessment of solid-state photophysical properties, please add fluorescence lifetime of the monomers and the macrocycles reported.

Response to question 1 of referee 3

According to the Reviewer’s advice, the fluorescence lifetime of the monomer and the macrocycle has been added. The fluorescence lifetimes of BT-M in the solid state were measured as 8.45 ns (Figure R37), whereas BT-LC showed relatively longer lifetimes of 11.25 ns (Figure R38). Please see Page 5 in revised manuscript and Figure S36 and S37 in revised Supplementary Information.

The following discussion was added to the revised manuscript in page 4:

“The time-resolved emission decay properties of BT-M and BT-LC in the solid state were also studied. The fluorescence lifetimes of BT-M were measured as 8.45 ns, whereas BT-LC showed relatively longer lifetimes of 11.25 ns (Supplementary Figure 36, 37).”

Figure R37 (Supplementary Figure 36) PL decay spectra of BT-M in solid state.

Figure R38 (Supplementary Figure 37) PL decay spectra of BT-LC in solid state.

Question 2

For the synthesis, are there any other cyclic oligomers such as tetramer as the by-product?

There are no any other cyclic oligomers. The following discussion was added to the revised manuscript in Page 3:

“No other cyclic oligomers such as tetramer and pentamer were observed.”

Question 3

In page 6, “BT-BC” should be BT-LC.

This was done.

Question 4

Please define the abbreviated terms when they appear for the first time. “Tol” or “TOL”? Please be consistent.

The abbreviation is defined as “TOL”. The “Tol” are replaced by “TOL”.

References

1. Wang, Y., Xu, K., Li, B., Cui, L., Li, J., Jia, X., Zhao, H., Fang, J., Li, C., Efficient Separation of cis- and trans-1,2-Dichloroethene Isomers by Adaptive Biphen[3]arene Crystals. *Angew. Chem. Int. Ed.* **58**, 10281-10284 (2019).
2. Xu, K., Zhang, Z., Yu, C., Wang, B., Dong, M., Zeng, X., Gou, R., Cui, L., Li, C. J. A Modular Synthetic Strategy for Functional Macrocycles. *Angew. Chem. Int. Ed.* **59**, 7214-7218 (2020).
3. (a) Perdew, J. P., Burke, K., Ernzerhof, M. Generalized gradient approximation made simple. *Phys. Rev. Lett.* **77**, 3865-3868 (1996). (b) Perdew, J. P., Burke, K., Ernzerhof, M. Errata: Generalized gradient approximation made simple. *Phys. Rev. Lett.* **78**, 1396-1396 (1997). (c) Adamo, C., Barone, V. Toward reliable density functional methods without adjustable parameters: The PBE0 model. *J. Chem. Phys.* **110**, 6158-6169 (1999).
4. Gaussian 16 revision A.03.
5. Dreuw, A., Head-Gordon, M. Single-Reference ab Initio Methods for the Calculation of Excited States of Large Molecules. *Chem. Rev.* **105**, 4009-4037 (2005).
6. Matsika, S. Electronic Structure Methods for the Description of Nonadiabatic Effects and Conical Intersections. *Chem. Rev.* **121**, 9407-9449 (2021).
7. Koga, N., Morokuma, K. Determination of the lowest energy point on the crossing seam between two potential surfaces using the energy gradient. *Chem. Phys. Lett.* **119**, 371-374 (1985).
8. (a) Zhao, H., Bian, W., Liu, K. A theoretical study of the reaction of O(³P) with isobutene. *J. Phys. Chem. A* **110**, 7858-7866 (2006). (b) Zhao, S., Wu, W., Zhao, H., Wang, H., Yang, C., Liu, K., Su, H. Adiabatic and nonadiabatic reaction pathways of the O(³P) with propyne. *J. Phys. Chem. A* **113**, 23-34 (2009). (c) Liu, K., Li, Y., Su, J., Wang, B. The reliability of DFT methods to predict electronic structures and minimum energy crossing point for [Fe^{IV}O](OH)₂ models: A comparison study with MCQDPT method. *J. Comput. Chem.* **35**, 703-710 (2014). (d) Li, H., Li, D., Zeng, X., Liu, K., Beckers, H., Schaefer, H. F. III., Esselman, B. J., McMahan, R. J. Toward Understanding the Decomposition of Carbonyl Diazide (N₃)₂C=O and Formation of Diazirone cycl-N₂CO: Experiment and Computations *J. Phys. Chem. A* **119**, 8903-8911 (2015). (e) Wu, Z., Feng, R., Li, H., Xu, J., Deng, G., Abe, M., Begue, D., Liu, K., Zeng, X. Fast Heavy-Atom Tunneling in Trifluoroacetyl Nitrene. *Angew. Chem. Int. Ed.* **56**, 15672-15676 (2017).
9. CYLview20; Legault, C. Y., Université de Sherbrooke, 2020 (<http://www.cylview.org>)

10. Peng, Z., Zhang, K., Huang, Z., Wang, Z., Duttwyler, S., Wang, Y., Lu, P., Emissions from a triphenylamine–benzothiadiazole–monocarbaborane triad and its applications as a fluorescent chemosensor and a white OLED component. *J Mater Chem C*, **7**, 2430-2435 (2019).
11. Sun, X., Xu, X., Qiu, W., Yu, G., Zhang, H., Gao, X., Chen, S., Song, Y., Liu, Y., A non-planar pentaphenylbenzene functionalized benzo[2,1,3]thiadiazole derivative as a novel red molecular emitter for non-doped organic light-emitting diodes. *J Mater Chem*, **18**, 2709 (2008).
12. Zhao, Z., Deng, C., Chen, S., Lam, JWY., Qin, W., Lu, P., Wang, Z., Kwok, HS., Ma, Y., Qiu, H., Tang, BZ., Full emission color tuning in luminogens constructed from tetraphenylethene, benzo-2,1,3-thiadiazole and thiophene building blocks. *Chem Commun*, **47**, 8847-8849 (2011).
13. Lee, WWH., Zhao, Z., Cai, Y., Xu, Z., Yu, Y., Xiong, Y., Kwok, RTK., Chen, Y., Leung, NLC., Ma, D., Lam, JWY., Qin, A., Tang, BZ., Facile access to deep red/near-infrared emissive AIEgens for efficient non-doped OLEDs. *Chem Sci*, **9**, 6118-6125 (2018).
14. Li, Y., Wang, W., Zhuang, Z., Wang, Z., Lin, G., Shen, P., Chen, S., Zhao, Z., Tang, BZ., Efficient red AIEgens based on tetraphenylethene: synthesis, structure, photoluminescence and electroluminescence. *J Mater Chem C*, **6**, 5900-5907 (2018).
15. Thangthong, A., Prachumrak, N., Sudyoasuk, T., Namuangruk, S., Keawin, T., Jungstittiwong, S., Kungwan, N., Promarak, V., Multi-triphenylamine–functionalized dithienylbenzothiadiazoles as hole-transporting non-doped red emitters for efficient simple solution processed pure red organic light-emitting diodes. *Org Electron*, **21**, 117-125 (2015).
16. Angioni, E., Chapran, M., Ivaniuk, K., Kostiv, N., Cherpak, V., Stakhira, P., Lazauskas, A., Tamulevius, S., Volyniuk, D., Findlay, NJ., Tuttle, T., Grazulevicius, JV., Skabara, PJ., A single emitting layer white OLED based on exciplex interface emission. *J Mater Chem C*, **4**, 3851-3856 (2016).
17. Pathak, A., Justin Thomas, KR., Singh, M., Jou, JH., Fine-Tuning of Photophysical and Electroluminescence Properties of Benzothiadiazole-Based Emitters by Methyl Substitution. *J Org Chem*, **82**, 11512-11523 (2017).
18. Zhang, Y., Zhou, X., Zhou, C., Su, Q., Chen, S., Song, J., Wong, W.-Y., High-efficiency organic electroluminescent materials based on the D-A-D type with sterically hindered methyl groups. *J. Mater. Chem. C*, **8**, 6851-6860 (2020).

Reviewer #1 (Remarks to the Author):

Although the authors showed considerable efforts to address the reviewers' comments in their revised submission, this reviewer remains hesitating to recommend for publication in Nat. Comm., due to the following.

1. The main concept of "macrocyclization-induced emission enhancement (MIEE)" is still not fully validated. For comparison, the well known aggregation- or vibration-induced emissions focus on the fluorescence changes of same molecules under different states. This manuscript, however, is comparing fluorescence between different ones, namely macrocycles and their synthetic precursors (fractional parts). Therefore, although I consent with the experimental and computational results, the authors' comparisons seem somewhat unfair. What I am also wondering is the role of methylene linkers in the macrocycles which may cause certain molecular strains and "restrict intramolecular motions" as the authors mentioned. What if all or part of the methylenes are replaced by more flexible and extended linkers?

2. The OLED performance is not quite exciting. When talking about any types of well developed devices, readers of high-profile journals would expect to see something significant.

Reviewer #3 (Remarks to the Author):

Since reviewers' comments have been well addressed, the revised manuscript is recommended for publication.

To Referee 1:

We greatly appreciate the reviewer's very positive comments and useful advices. Herewith, we addressed the referee's comments as follows:

Question 1

The main concept of "macrocyclization-induced emission enhancement (MIEE)" is still not fully validated. For comparison, the well known aggregation- or vibration-induced emissions focus on the fluorescence changes of same molecules under different states. This manuscript, however, is comparing fluorescence between different ones, namely macrocycles and their synthetic precursors (fractional parts). Therefore, although I consent with the experimental and computational results, the authors' comparisons seem somewhat unfair. What I am also wondering is the role of methylene linkers in the macrocycles which may cause certain molecular strains and "restrict intramolecular motions" as the authors mentioned. What if all or part of the methylenes are replaced by more flexible and extended linkers?

Response to question 1 of referee 1

Thanks a lot for consenting with our experimental and computational results. This reviewer's previous great comments and advices, especially for the scope and mechanism of our strategy, help a lot for improving the quality of our manuscript. As a well-known strategy, aggregation-induced emission (AIE) provides an efficient approach to enhance emission and has been applied in many fields. Tetraphenyl ethylene (TPE) molecules are non-emissive in solution and highly emissive in aggregate (Figure R1). The essence is the restriction of intramolecular rotations of luminophore by aggregating. In most cases, the emission performance is compared between free single luminogen with aggregates which is composed of lots of luminogens. As shown in Figure R1, our MIEE strategy is also to restrict intramolecular motion of luminophore. The difference is utilizing the skeleton of macrocycle but not aggregating. The linker of methylene played the role of locking luminophores and would not disturb the emission of luminophores due to its

non-conjugated character. Moreover, the computational results gave a reasonable mechanism. With the double-bonding tendency, the non-radiative relaxation process requires that the torsion angle between benzothiadiazole and benzene ring can be twisted to near 20 degrees at $\text{MECP}_{\text{S1/S0}}$. That is to say, benzothiadiazole and benzene ring tend to be in the same plane at $\text{MECP}_{\text{S1/S0}}$. Unlike BT-M, BT-LC has a rigid triangular geometry and the rotation of the corresponding torsion angle would be limited (Figure 3a and S48). Therefore, BT-LC can avoid the process of $\text{MECP}_{\text{S1/S0}}$ non-radiative relaxation and its fluorescence efficiency is enhanced.

Considering that AIE is based on the aggregation of luminogens and our strategy is due to the cyclization of a few of luminogens by methylenes, we think “macrocyclization-induced emission enhancement (MIEE)” is an appropriate term. The following description was added to the introduction part of the revised manuscript: *“Since the emission enhancement is due to the cyclization of a few of luminogens by methylenes, it is termed as macrocyclization-induced emission enhancement (MIEE) (Scheme S1).”* And the following figure was added to the supporting information (Scheme S1).

Figure R1 (Scheme S1). Illustration of aggregation-induced emission (AIE) and macrocyclization-induced emission enhancement (MIEE).

It is an excellent suggestion to replace all or part of the methylenes by more flexible and extended linkers. However, such synthesis is extremely difficult and beyond our ability. Actually, macrocycles with functionalized skeletons are hard to synthesize, although functional substituents can be conveniently attached to the macrocycle periphery. Our recent work developed a general and modular method to customize functional macrocycles containing diverse skeletons that are connected by methylenes. This modular synthesis provides the convenient production of diverse fluorescence macrocycles, ensuring the availability and versatility of MIEE. However, we are very sorry that the methylenes cannot be replaced by more flexible and extended linkers in this system.

Question 2

The OLED performance is not quite exciting. When talking about any types of well developed devices, readers of high-profile journals would expect to see something significant.

Response to question 2 of referee 1

Certainly, compared with reported BT-based emitters, macrocycle BT-LC showed medium current efficiency, power efficiency and external quantum efficiency. Although the performance of the device is inferior to that of the current state-of-art, it is the first example of macrocyclic arene-based OLED. Macrocycle would be potentially applied in OLEDs considering the following two points: 1) our modular synthesis method could conveniently produce diverse fluorescence macrocycles; 2) MIEE is an efficient strategy to improve Φ_{PL} values of chromophores.” From this point, this work is believed to be merited in this high-profile journal.